# LATENT SPACE STRUCTURING FOR CONDITIONAL TABULAR DATA GENERATION ON IMBALANCED DATASETS

## ABSTRACT

Generating synthetic tabular data under severe class imbalance is essential for domains where rare but high-impact events drive decision-making. However, most generative models either overlook minority groups or fail to produce samples that are useful for downstream learning. We introduce CTTVAE, a Conditional Transformer-based Tabular Variational Autoencoder equipped with two complementary mechanisms: (i) a class-aware triplet margin loss that restructures the latent space for sharper intra-class compactness and inter-class separation, and (ii) a training-by-sampling strategy that adaptively increases exposure to underrepresented groups. Together, these components form CTTVAE+TBS, a framework that consistently yields more representative and utility-aligned samples without destabilizing training. Across six real-world benchmarks, CTTVAE+TBS achieves the strongest downstream utility on minority classes, often surpassing models trained on the original imbalanced data while maintaining competitive fidelity and bridging the gap for privacy for interpolation-based sampling methods and deep generative methods. Ablation studies further confirm that both latent structuring and targeted sampling contribute to these gains. By explicitly prioritizing downstream performance in rare categories, CTTVAE+TBS provides a robust and interpretable solution for conditional tabular data generation, with direct applicability to industries such as healthcare, fraud detection, and predictive maintenance where even small gains in minority cases can be critical.

## 1 INTRODUCTION

Generating high-quality synthetic tabular data has become increasingly important for addressing challenges such as data scarcity, privacy constraints Borisov et al. (2022), and class imbalance. These issues are particularly critical in domains like healthcare Hernandez et al. (2022), fraud detection, and industrial monitoring, where rare but high-impact events, such as disease diagnosis, fraudulent transactions, or equipment failures, are severely underrepresented. Models trained on such imbalanced datasets often fail to capture meaningful minority-class patterns, leading to biased predictions and poor generalization D'souza et al. (2025). Given the ubiquity of tabular data, improving synthetic generation for downstream learning is a pressing need James et al. (2021).

Classical oversampling methods such as SMOTE Chawla et al. (2002) remain popular due to their simplicity, but they only interpolate between input-space samples and often yield unrealistic data in high dimensions Batista et al. (2004). Deep generative models (VAEs, GANs, and diffusion models) provide more expressive alternatives. Transformer-based VAEs Wang & Nguyen (2025) leverage self-attention to capture rich inter-feature dependencies, but they typically struggle with severe imbalance, producing poor-quality minority samples in low-density regions D'souza et al. (2025). Thus, two challenges remain: (i) generative models tend to overlook rare categories unless explicitly conditioned or regularized, and (ii) minority examples require latent representations that are both expressive and class-discriminative.

We propose the Conditional Transformer-based Tabular Variational Autoencoder (CTTVAE), a framework that combines latent space structuring with adaptive sampling to explicitly address class imbalance. CTTVAE incorporates a class-aware triplet margin loss to promote intra-class compactness and inter-class separation, and integrates a training-by-sampling (TBS) strategy that increases exposure to underrepresented groups, which will be referred as CTTVAE+TBS. Together,

these mechanisms enable conditional generation that is both representative and utility-aligned, particularly for minority categories. Unlike interpolation methods, CTTVAE operates in a structured latent space, producing semantically coherent samples without sacrificing training stability.

We evaluate CTTVAE across six public benchmarks, comparing it against one classical interpolation baseline and six generative models. Our study provides a systematic analysis of fidelity, privacy, and downstream utility Alaa et al. (2022), and includes ablation experiments isolating the contributions of latent structuring and sampling. Results show that CTTVAE significantly improves downstream utility on minority classes while maintaining competitive fidelity and privacy preservation.

The key contributions of this work are:

1. A conditional transformer-based VAE that explicitly improves minority-class utility through latent space structuring and targeted sampling.

2. Unlike prior models that either interpolate blindly in the input space or regularize the latent space without task awareness, CTTVAE explicitly restructures the latent manifold to reflect class semantics while simultaneously balancing exposure to rare groups.

3. A dual structuring that yields a controllable and general framework and extends naturally to any categorical conditioning variable, far beyond binary class imbalance.

4. Through extensive evaluation across six benchmarks, we demonstrate that CTTVAE consistently improves minority-class utility and privacy.

## 2 RELATED WORK

### 2.1 INTERPOLATION METHODS

Traditional oversampling techniques serve as strong baselines for handling class imbalance. The Synthetic Minority Over-sampling Technique (SMOTE) Chawla et al. (2002) generates synthetic examples by linearly interpolating between minority class samples. Despite lacking the sophistication of deep models, this method can perform surprisingly well in combination with robust classifiers.

### 2.2 DEEP GENERATIVE MODELS

Generative models for tabular data have emerged as powerful tools for addressing challenges such as data scarcity, privacy preservation, and class imbalance. Most high-performing models come from the 3 main generative model families: Variational Autoencoders (VAEs), Generative Adversarial Networks (GANs), Diffusion models Kingma et al. (2013); Goodfellow et al. (2014); Ho et al. (2020). Among the early works in this area, CTGAN and TVAE Xu et al. (2019) introduced deep generative modeling frameworks specifically tailored to the tabular setting. CTGAN uses a conditional GAN architecture combined with mode-specific normalization to model mixed-type features and imbalanced class distributions, while TVAE formulates generation as a variational inference problem, enabling probabilistic modeling of heterogeneous feature types.

To improve the synthesis of mixed-type tabular data, CTAB-GAN Zhao et al. (2021) extends conditional GANs by introducing classification loss for better supervision, type-specific encoding for continuous and categorical variables, and lightweight preprocessing to handle long-tailed continuous distributions. Its design increases robustness to class imbalance and skewed data distributions. CopulaGAN, introduced in the SDV opensource library Patki et al. (2016), enhances CTGAN by combining it with a Gaussian copula-based normalization procedure.

Other recent methods such as Overlap Region Detection (ORD) D'souza et al. (2025) have shown that data imbalance often leads to poor generalization due to decision boundaries being dominated by majority-class instances. ORD addresses by selectively increasing the density of minority class data in critical regions of the data space, thereby improving classifier performance. Their results suggest that explicitly shaping the distribution of training samples can substantially enhance downstream utility, especially for underrepresented classes.

Recently, TabDDPM Kotelnikov et al. (2023) introduced diffusion-based generative modeling to the tabular domain, leveraging iterative denoising processes to achieve high-fidelity and privacy-aware samples. While TabDDPM reports state-of-the-art performance on several fidelity benchmarks, it

does not support conditional generation by design. TabDiff Shi et al. (2025) models tabular data with a continuous-time diffusion process over mixed numerical and categorical features, incorporating learnable per-feature noise schedules.

Several other models have also been proposed for tabular data generation, including CTAB-GAN+ Zhao et al. (2024), TabSyn Zhang et al. (2023), MedGAN Choi et al. (2017), and STaSy Kim et al. (2022), among others. All these methods highlight progress in realistic tabular generation, yet few tackle conditional synthesis under severe class imbalance.

## 3 METHODS

Our goal is to design a generative framework that explicitly improves the downstream utility of synthetic tabular data in imbalanced settings, with a particular focus on minority classes. To this end, we build on the TTVAE model and introduce CTTVAE+TBS, which combines latent space structuring with adaptive sampling.

### 3.1 OVERVIEW OF TTVAE

TTVAE is a generative model for tabular data that extends the VAE framework by leveraging the Transformer's Vaswani et al. (2017) capabilities for heterogeneous tabular features Wang & Nguyen (2025). A Transformer-based encoder produces contextualized embeddings Huang et al. (2020), denoted $\mathbf{h}$, which capture both local and global dependencies between features. These embeddings allow the model to represent inter-feature relationships in a compressed format and seamlessly integrate categorical (one-hot encoded) and numerical (modeled through a Variational Gaussian Mixture) variables. Given an input $\mathbf{x}$, the encoder outputs:

$$\mathbf{h} = f_{\text{enc}}^{\text{Transf}}(\mathbf{x}), \quad \mathbf{z} \sim q_\phi(\mathbf{z}|\mathbf{x}), \tag{1}$$

where $\mathbf{h}$ captures inter-feature dependencies and $\mathbf{z}$ is sampled from the variational posterior. The decoder reconstructs $\mathbf{x}$ using both:

$$\hat{\mathbf{x}} \sim p_\theta(\mathbf{x}|\mathbf{z}, \mathbf{h}). \tag{2}$$

Instead of the standard KL divergence term, TTVAE applies a Maximum Mean Discrepancy (MMD) penalty Gretton et al. (2012) between the aggregated posterior $q(\mathbf{z})$ and the Gaussian prior $p(\mathbf{z})$, yielding the objective:

$$\mathcal{L}_{\text{TTVAE}} = -\mathbb{E}_{q_\phi(\mathbf{z}|\mathbf{x})}[\log p_\theta(\mathbf{x}|\mathbf{z}, \mathbf{h})] + \beta \cdot \text{MMD}(q(\mathbf{z}), p(\mathbf{z})), \tag{3}$$

where $\beta$ controls the intensity of the MMD term. This formulation encourages a well-regularized latent space that captures higher-order moments and supports interpolation-based sampling. During generation, synthetic latent vectors are created via triangular interpolation over real latent encodings Fonseca & Bacao (2023), inspired by latent mixup Beckham et al. (2019), to promote semantic coherence and improve sample realism.

While TTVAE effectively models complex tabular structures, it lacks mechanisms to explicitly organize the latent space with respect to class information. As a result, it may struggle to generate useful samples for underrepresented classes when interpolation crosses ambiguous or low-density regions. This limitation motivates the need for class-aware latent structuring introduced in CTTVAE.

### 3.2 CTTVAE

As the first component of our proposed framework CTTVAE+TBS, CTTVAE extends TTVAE to structure the latent space with respect to class information while keeping the same transformer architecture. However, it is not inherently designed to prioritize or structure the latent space with respect to class or category-level semantics. This can limit their ability to generate useful samples for underrepresented groups, especially when generating data in ambiguous regions of the latent space. In comparison to ORD which operates in the data space, our approach takes a different perspective by directly structuring the latent space during training to encode class-aware relationships, enabling more reliable and controllable generation and improving sample quality of underrepresented classes.

To address this, we enhance the latent space geometry by incorporating triplet loss as it has proven to effectively work for VAEs Ishfaq et al. (2018), specifically we implement the **triplet margin loss**.

This addition encourages latent representations of instances from the same class to be embedded closely, while pushing apart samples from different classes. It directly acts on the mean latent vectors of the encoder.

Let $\mathbf{z}_a$ be the latent encoding of an anchor instance, $\mathbf{z}_p$ a positive sample from the same class, and $\mathbf{z}_n$ a negative sample from a different class. The triplet margin loss is defined as:

$$\mathcal{L}_{\text{triplet}} = \sum_{(a,p,n) \in \mathcal{T}_{\text{batch}}} \max\left( \|\mathbf{z}_a - \mathbf{z}_p\|_2^2 - \|\mathbf{z}_a - \mathbf{z}_n\|_2^2 + m, \ 0 \right), \tag{4}$$

where $\mathcal{T}_{\text{batch}}$ denotes the set of all valid triplets in the mini-batch and $m$ is the margin hyperparameter. This objective encourages embeddings of the same class to lie closer together than those of different classes by at least margin $m$. We adopt **semi-hard negative mining**, following Schroff et al. (2015), to guide the model towards informative comparisons, selecting $\mathbf{z}_n$ such that:

$$\|\mathbf{z}_a - \mathbf{z}_p\|_2^2 < \|\mathbf{z}_a - \mathbf{z}_n\|_2^2 < \|\mathbf{z}_a - \mathbf{z}_p\|_2^2 + m \tag{5}$$

The detailed procedure is found in Algorithm 1. The final training objective combines the TTVAE loss with the triplet margin loss:

$$\mathcal{L}_{\text{CTTVAE}} = -\mathbb{E}_{q_\phi(\mathbf{z}|\mathbf{x})}[\log p_\theta(\mathbf{x}|\mathbf{z}, \mathbf{h})] + \beta \cdot \text{MMD}(q(\mathbf{z}), p(\mathbf{z})) + \alpha \cdot \mathcal{L}_{\text{triplet}}$$

where $\mathbf{x}$ is the input data, $\mathbf{h}$ is the contextual embedding produced by the Transformer encoder to capture inter-feature dependencies, and $\mathbf{z}$ is the latent representation sampled from the approximate posterior $q_\phi(\mathbf{z}|\mathbf{x})$. The term $\mathbb{E}_{q_\phi(\mathbf{z}|\mathbf{x})}[\log p_\theta(\mathbf{x}|\mathbf{z}, \mathbf{h})]$ represents the reconstruction loss. The term $\text{MMD}(q(\mathbf{z}), p(\mathbf{z}))$ represents the MMD loss. The hyperparameters $\beta$ and $\alpha$ control the degree of intensity of the MMD term and the triplet loss term respectively. A higher $\beta$ promotes a more distentangled latent space and a lower one improves the reconstruction loss. Increasing $\alpha$ tightens intra-class clustering and widens inter-class separation.

This leads to a latent space that is better aligned with the desired class label eliminating the blending of unrelated samples(see Figure 7 in Appendix F). Furthermore, our framework allows the user to specify any categorical feature during training instead of class variable. This flexibility is especially valuable in use cases where the downstream task depends on factors other than the class label, such as demographic group, region, or product type.

**Conditional Generation**    CTTVAE performs class-conditional generation by interpolating only within class-specific latent subsets (Figure 1). The encoder outputs $(\mu_i, \sigma_i, h_i)$ for each input $x_i$, and we then draw $z_i \sim \mathcal{N}(\mu_i, \text{diag}(\sigma_i^2))$.

For a target class $c$, we retain the subset $\mathcal{S}_c = \{(z_i, h_i) : y_i = c\}$. For each randomly chosen base point $z_i \in \mathcal{S}_c$, we build its $k$-nearest-neighbor set $\mathcal{N}_k(z_i)$ (Minkowski metric; neighbors ranked by increasing distance). Denote the $r$-th neighbor by $\nu_{i,r}$. The so-called *triangle* interpolation (as implemented in the TTVAE code) generalizes the original 3-point triangular interpolation to the $k$-NN case. A synthetic latent point is obtained by sampling within the local convex region around $z_i$ using inverse-rank weights and per-neighbor random scalars:

$$w_r = \frac{k - r}{\frac{k(k-1)}{2}} \quad (r = 1, \dots, k), \qquad u_r \sim \mathcal{U}(0,1), \qquad \hat{z} = z_i + \sum_{r=1}^{k} w_r\, u_r\, (\nu_{i,r} - z_i). \tag{6}$$

The decoder receives both the synthetic latent vectors and the filtered encoder outputs and reconstructs $\hat{x} \sim p_\theta(x \mid \hat{z}, h)$. This ensures that generation remains confined to a coherent latent region aligned with the target class. Additionally, $u_r$ provides random scaling to diversify interpolated samples.

This approach eliminates the need for a conditioning network, instead relying on the structurally aligned latent space learned during training. Since interpolation occurs within condition-specific regions, generated samples preserve class semantics and avoid blending across categories D'souza et al. (2025). While the conditioning mechanism is generalizable to any discrete feature, in this work we focus on the class label, as improving minority-class utility is our primary objective.

Our framework establishes a new paradigm in which the latent space is intentionally restructured for task relevance while the training process is guided to preserve minority representation. This coupling of geometric structuring and sampling control creates a generative framework explicitly tailored to imbalanced tabular learning, setting it apart from existing methods that either ignore class structure or rely on naive interpolation.

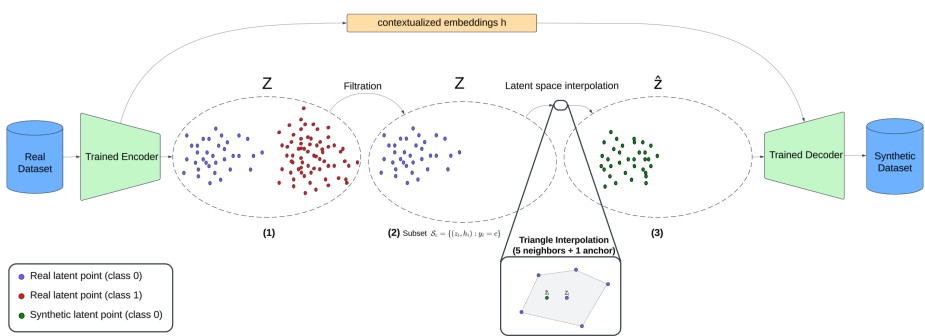

Figure 1: Conditional generation with CTTVAE. **(1)** The trained encoder maps real samples to latent representations $\mathbf{z}$, from which class-specific subsets are retained. **(2)** A filtration step isolates the latent region corresponding to the target class ($\mathcal{S}_c$). **(3)** Synthetic latent points $\hat{\mathbf{z}}$ are generated by local triangle interpolation: for each anchor latent point $z_i$, sampling occurs within the convex region spanned by its $k = 5$ nearest neighbors (bottom zoom panel). The decoder then reconstructs synthetic samples from $(\hat{\mathbf{z}}, \mathbf{h})$.

### 3.3 TRAINING-BY-SAMPLING (TBS)

Our second component, TBS, is a batch sampling strategy introduced in CTGAN Xu et al. (2019) to mitigate representation bias in tabular datasets, particularly when categorical features exhibit strong imbalance. Rather than drawing training batches uniformly at random, TBS constructs each batch by repeatedly selecting a specific value in a discrete column and sampling data points matching that value. This process ensures that all discrete values across all columns are regularly seen during training, even if their marginal frequency in the dataset is low.

We adopt a variant of the TBS concept, where sampling is guided solely by a user-specified categorical feature rather than sampling over all discrete columns. We do it on only the class label to address the imbalance to have a smoothed class sampling distribution. Specifically, we form a convex combination between the original class distribution $P_{\text{orig}}$ and the uniform distribution $P_{\text{uniform}}$. The resulting sampling probability mass function (PMF) for each class $c$ is given by:

$$\text{PMF}[c] = \lambda \cdot P_{\text{orig}}[c] + (1 - \lambda) \cdot P_{\text{uniform}}[c], \tag{7}$$

where $\lambda \in [0, 1]$ is a tunable hyperparameter. $\lambda = 1$ samples from the original class proportions, while $\lambda = 0$ does uniform sampling. Intermediate values offer a trade-off that improves exposure to rare classes without discarding the underlying data distribution, to mitigate risks of overfitting to the minority class.

## 4 RESULTS

**Datasets** We extensively evaluate our methods against existing alternatives across various datasets with binary target variables, different properties, sizes, and number of features to evaluate models in different real-world settings, as seen in Table 1. The first three datasets have been used in most of the literature regarding tabular data generation, the other three have been chosen to explore more extreme cases.

**Experiments** Each dataset is processed independently by each method to generate fully synthetic data that reflects the training distribution. The training and test subsets are split to have the same imbalance ratio as the full dataset. We did 25 hyperparameter tuning trials for all generative methods (see appendix G for more details), including the $\alpha$ and $\beta$ hyperparameters from our loss function. We run our methods on A100 GPUs.

Table 1: Summary of the datasets used in our experiments. CH = Churn Modeling, AD = Adult, DE = Default of Credit Card Clients, CR = Credit Card Fraud Detection (50k instances - due to limited resources, we undersampled the majority class while keeping the same number of minority instances), MA = Machine Predictive Maintenance, VE = Vehicle Insurance Claims. "IR" denotes the imbalance ratio between the majority and minority class in the training set.

| Abbr. | Train/Test | #Num. Features | #Cat. Features | Target Column | IR |
|-------|-----------|----------------|----------------|---------------|-----|
| CH | 8k / 2k | 6 | 4 | "Exited" | 3.9 |
| AD | 24,111 / 6,028 | 6 | 8 | "income" | 3.0 |
| DE | 24k / 6k | 20 | 3 | "default.payment.next.month" | 3.5 |
| CR | 40,378 / 10,095 | 29 | 0 | "Class" | 105.7 |
| MA | 8k / 2k | 5 | 1 | "Target" | 28.5 |
| VE | 12,080 / 3,020 | 1 | 29 | "FraudFound_P" | 15.9 |

## 4.1 Utility Scores

To assess the utility of the synthetic data for downstream tasks, we employ the Machine Learning Efficacy (MLE) score. It evaluates the similarity in classification performance when models are trained on synthetic data and tested on real data, compared to models trained and tested entirely on real data. We compute the average F1 score using CatBoost Prokhorenkova et al. (2018), averaging results over three independent generations for each method and dataset. A higher MLE indicates better alignment with real-data performance, suggesting greater practical utility of the synthetic data.

Table 2 summarizes the results for minority classes (and Table 6 in the appendix for both). Across all datasets, our method achieves consistently the top 2 MLE scores on all datasets. In particular, it has the best result for 5 out of 6 datasets and the 2nd best for the remaining one. We outperform other generative models by significant margins, especially for highly imbalanced datasets (up to 13.7x better for VE, up to 2x for CR and up to 3.5x for MA). In comparison with TTVAE, our extension significantly improves performance on all datasets. These improvements are obtained while keeping majority-class performance stable, which is the intended behavior for oversampling in imbalanced regimes. SMOTE remains a strong baseline for utility and outperforms most models in other papers as well however, it lacks the ability to scale to high-dimensional data, provide no privacy safeguards, and cannot handle flexible conditional generation (Table 4). CTTVAE addresses all three, showing why deep generative models are essential in practice despite surface-level similarity in some scores.

Table 2: Average MLE and standard deviation over three generations computed with CatBoost across datasets for *minority samples only*. **Bold** represents the best results on each dataset and underlined represents the second best results on each dataset. The performances on the majority class remains stable for all the considered methods. "Real" represents the scores trained on the original dataset. Higher means better.

| Method | CH | AD | DE | CR | MA | VE |
|--------|-----|-----|-----|-----|-----|-----|
| Real | 0.607±0.001 | 0.728±0.002 | 0.468±0.003 | 0.893±0.001 | 0.790±0.004 | 0.112±0.002 |
| CTGAN | 0.559±0.042 | 0.677±0.001 | 0.459±0.020 | 0.428±0.161 | 0.327±0.010 | 0.011±0.010 |
| TVAE | 0.502±0.015 | 0.609±0.003 | 0.397±0.006 | 0.838±0.020 | 0.189±0.006 | 0.001±0.001 |
| CopulaGAN | 0.560±0.010 | 0.569±0.004 | 0.474±0.038 | 0.450±0.190 | 0.302±0.023 | 0.053±0.060 |
| CTABGAN | 0.575±0.020 | 0.612±0.002 | 0.466±0.039 | 0.498±0.172 | 0.327±0.006 | 0.071±0.012 |
| TabDiff | 0.574±0.010 | 0.679±0.002 | 0.478±0.001 | 0.869±0.012. | 0.628±0.083 | 0.071±0.006 |
| SMOTE | 0.608±0.014 | 0.694±0.001 | 0.501±0.001 | **0.891±0.001** | **0.678±0.035** | 0.113±0.018 |
| TTVAE | 0.607±0.004 | 0.689±0.001 | 0.463±0.004 | 0.857±0.012 | 0.560±0.017 | 0.072±0.002 |
| CTTVAE+TBS | **0.628±0.006** | **0.703±0.002** | **0.512±0.009** | 0.881±0.004 | **0.684±0.045** | **0.137±0.016** |

## 4.2 Fidelity Analysis

We evaluate the fidelity of the synthetic data using three metrics: Wasserstein Distance (WD), Jensen–Shannon Divergence (JSD), and pairwise correlation error (see appendix A.2 for details).

Table 3 shows that diffusion and interpolation-based sampling methods yield on average the strongest fidelity scores overall. SMOTE achieves the lowest WD, JSD, and correlation error which is expected since interpolated samples remain very close to existing records. Among deep generative models, TabDiff obtains the lowest WD and JSD with TTVAE close behind. However, CTTVAE+TBS is comparable on most fidelity metrics, while offering the minority-class utility gains absent from TTVAE and TabDiff. Correlation error further highlights this balance with CTTVAE+TBS achieving errors slightly lower than TTVAE (2.11% vs. 2.14%) and around the same as TabDiff (2.10% ). It is also substantially lower than GAN-based methods (6–12%). Interpolation-based generative methods show more stability than the other deep generative models. Tables 8-10 show results per dataset.

Table 3: Average WD and JSD and standard deviation (per class), and average pairwise correlation error and standard deviation (%) over all datasets. **Bold** and underline indicate best and second-best results respectively. Lower means better. Maj = majority class, Min = minority class, Avg = average.

| Method | WD $\downarrow$ | | JSD $\downarrow$ | | Corr. (%) $\downarrow$ |
|---|---|---|---|---|---|
| | Maj. | Min. | Maj. | Min. | Avg. |
| CTGAN | 0.103±0.026 | 0.128±0.037 | 0.084±0.039 | 0.092±0.045 | 11.48±12.75 |
| TVAE | 0.135±0.067 | 0.272±0.064 | 0.141±0.053 | 0.178±0.088 | 6.46±2.97 |
| CopulaGAN | 0.123±0.038 | 0.167±0.056 | 0.092±0.037 | 0.100±0.042 | 12.81±13.50 |
| CTABGAN | 0.159±0.096 | 0.205±0.084 | 0.076±0.045 | 0.078±0.051 | 6.22±2.18 |
| TabDiff | **0.030±0.006** | 0.069±0.042 | 0.027±0.011 | 0.034±0.017 | 2.10±0.73 |
| SMOTE | 0.031±0.014 | **0.056±0.015** | **0.009±0.007** | **0.019±0.015** | **1.43±0.31** |
| TTVAE | 0.057±0.048 | 0.111±0.071 | 0.028±0.013 | 0.044±0.017 | 2.14±1.57 |
| CTTVAE+TBS | 0.065±0.040 | 0.093±0.033 | 0.035±0.023 | 0.048±0.019 | 2.11±1.60 |

Figure 2 supports these findings: CTTVAE, TTVAE, TabDiff and SMOTE consistently display the lightest heatmaps, indicating minimal deviation from the true correlation structure. In contrast, TVAE, CTGAN and CTABGAN show heavier distortions, confirming their higher correlation errors. These findings show that CTTVAE provides a strong fidelity–utility trade-off, maintaining competitive fidelity among generative models while significantly outperforming them on minority utility.

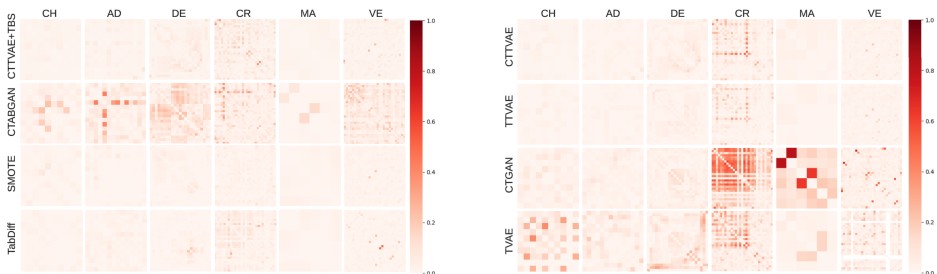

Figure 2: The absolute difference between correlation matrices computed on real and synthetic datasets. More intense red color indicates higher difference. Overall, TabDiff and our methods capture correlations better.

### 4.3 PRIVACY PRESERVATION

To evaluate potential privacy risks in the generated data, we rely on two Euclidean distance-based measures that focus on the proximity between synthetic and real samples. The Distance to Closest Record (DCR) quantifies the minimum distance from each synthetic record to its nearest real counterpart. Lower DCR values suggest a higher risk of memorization and worse privacy preservation. Complementing this, the Nearest Neighbour Distance Ratio (NNDR) assesses how distinct a synthetic point is by comparing the distance to its 2 closest real neighbors. If the ratio is near one, the synthetic point is similarly distant from multiple real records, reducing the likelihood that it mimics any single example. We report the 5[th] percentile to follow the precedent established in prior work such as CTABGAN Zhao et al. (2021).

Table 4 compares CTTVAE+TBS against interpolation baselines, since interpolation directly biases these distance metrics. Tables 11-13 show results per-dataset for all methods. As expected, SMOTE exhibits the weakest privacy, nearly two times worse than our method for minority samples, because convex combinations place synthetic points almost on top of real records. TTVAE has better privacy but CTTVAE+TBS achieves a clear margin with TTVAE and SMOTE for all classes and privacy metrics. With this we can deduce that latent-space restructuring combined with targeted sampling yields substantially stronger safeguards against memorization.

Full results against all other generative models are reported in Appendix Table 11. While some models report higher raw DCR values, this often reflects excessive drift away from real distributions, which correlates with poor utility and fidelity. By contrast, CTTVAE offers a balanced trade-off, maintaining strong privacy while clearly outperforming baselines on minority-class utility.

Table 4: Per-class Distance to Closest Record (DCR) and Nearest Neighbour Distance Ratio (NNDR) average across all datasets. **Bold** represents the best results and underline represents the second best on each metric. Higher values indicate better privacy. The standard deviations are higher due to the extremely imbalanced datasets (CR, MA, VE) distorting results for DCR. However, for NNDR, we see more stability as expected. Maj = majority class, Min = minority class.

| Method | DCR ↑ | | NNDR ↑ | |
|---|---|---|---|---|
| | Maj. | Min. | Maj. | Min. |
| SMOTE | 0.380±0.117 | 0.864±0.557 | 0.282±0.132 | 0.372±0.161 |
| TTVAE | 0.699±0.716 | 1.382±1.351 | 0.368±0.247 | 0.440±0.168 |
| CTTVAE+TBS | **1.587±2.254** | **1.528±1.698** | **0.534±0.186** | **0.543±0.139** |

## 4.4 ABLATION STUDY

We conduct an ablation study to disentangle the contributions of the triplet loss and the TBS strategy. Table 5 reports results relative to TTVAE across all datasets.

First, adding triplet loss (CTTVAE vs. TTVAE) yields consistent gains in minority utility (+0.032 on average) while maintaining stable performance on majority classes which shows that restructuring the latent space toward class separation produces more task-relevant minority samples. Importantly, CTTVAE also improves privacy with a much higher DCR/NNDR which reduces the risk of generating records overly close to real samples.

Second, incorporating TBS further amplifies these effects. CTTVAE+TBS achieves the largest overall gains on minority utility (+0.048), while keeping majority performance nearly unchanged. TBS also strengthens privacy across both majority and minority classes and, despite minor fluctuations, preserves fidelity at a competitive level. Figure 3 shows that while majority-class performance is stable across $\lambda$ values, minority-class scores benefit substantially from balanced sampling, highlighting the importance of controlled exposure.

The results of the ablation study further demonstrate that triplet loss improves minority class alignment in the latent space, while TBS provides robust training dynamics, and that their combination produces the best trade-off between utility, fidelity, and privacy.

Table 5: Ablation study results relative to TTVAE across all datasets. Higher is better for MLE, DCR, NNDR; lower is better for WD, JSD. **Bold** represents the best result and underline represents the second best result. Maj = majority class, Min = minority class, Avg = average.

| Method | Avg. MLE ↑ | | Avg. WD ↓ | | Avg. JSD ↓ | | DCR ↑ | | NNDR ↑ | | Corr. (%) ↓ |
|---|---|---|---|---|---|---|---|---|---|---|---|
| | Maj. | Min. | Maj. | Min. | Maj. | Min. | Maj. | Min. | Maj. | Min. | |
| TTVAE+TBS | **0** | +0.030 | +0.017 | _0.009_ | +0.013 | +0.006 | +0.075 | –0.112 | +0.025 | +0.036 | +0.24 |
| CTTVAE | -0.002 | _+0.032_ | **+0.005** | +0.010 | _+0.008_ | **–0.001** | **+0.888** | **+0.451** | _+0.161_ | +0.088 | +0.21 |
| CTTVAE+TBS | _-0.001_ | **+0.048** | +0.008 | **–0.018** | **+0.007** | _+0.004_ | **+0.888** | _+0.145_ | **+0.166** | **+0.103** | **–0.03** |

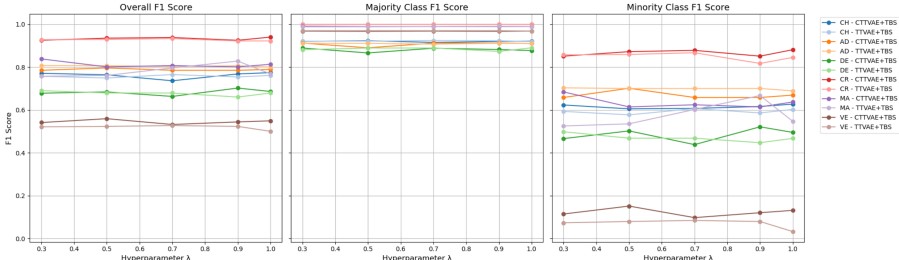

Figure 3: Impact on the minority class of the sampling hyperparameter $\lambda$ on F1 scores across datasets for CTTVAE+TBS and TTVAE+TBS. $\lambda = 1$ represents the models performances without aplying TBS. Performance on minority classes depends greatly on its value.

## 5 LIMITATIONS AND DISCUSSION

Our framework demonstrates consistent utility improvements across all datasets with strong gains for minority classes showing that structuring the latent space with triplet loss and balancing exposure through TBS are effective strategies for generating task-relevant data under imbalance. Importantly, these benefits come without degrading majority-class performance, which makes the method particularly suitable for domains where minority events drive downstream decisions. We also observe that combining TBS with a structured latent space consistently yields better performance, as seen with CTTVAE+TBS, compared to using TBS with an unstructured latent space (TTVAE+TBS).

Some trade-offs remain. The triplet loss adds computational overhead, which may limit scalability to very large datasets. Fidelity metrics also show that class-aware interpolation can underperform raw TTVAE in distributional alignment, while privacy scores indicate that interpolation-based models inherently place synthetic samples closer to real points. However, these effects are moderate, and the addition of TBS mitigates them by reducing overfitting and improving privacy without destabilizing training. Crucially, in imbalanced learning scenarios, slightly lower fidelity is an acceptable compromise when it yields substantially higher utility and stronger privacy protection. The practical value of synthetic data lies in improving downstream task performance while avoiding direct memorization. We argue this trade-off is not a drawback since this is more valuable for downstream deployment, where the goal is robust minority-class decision making rather than pixel-perfect distribution matching. This behavior is also reflected in Figure 5, where our methods attain higher minority-class utility without disproportionately sacrificing privacy or fidelity.

## 6 CONCLUSION

We introduced CTTVAE, a conditional transformer-based VAE that shows the impact of structuring data can have. It establishes a new paradigm for imbalanced tabular data generation by restructuring the latent space and guiding training to preserve minority representation. This structuring and adaptive sampling yields consistent improvements in downstream utility for rare classes while also enhancing privacy and keeping fidelity competitive. Unlike interpolation baselines that appear strong only because they produce samples close to real records, CTTVAE+TBS achieves a more meaningful balance, generating diverse, task-relevant and privacy-preserving data. These properties make it a practical solution for real-world domains such as fraud detection, predictive maintenance, and healthcare, where minority utility and privacy protection are paramount.

## 7 FUTURE WORK

While this study confirms the effectiveness of structuring latent spaces and sampling bias, several avenues remain open. TBS improves performance but requires tuning its hyperparameter $\lambda$. A natural extension of this work involves exploring more self-adaptive sampling strategies to optimize class exposure dynamically based on training dynamics or dataset properties, reducing manual intervention while preserving performance gains. Additionally, extending the privacy evaluation with metrics such as Membership Inference Attack Accuracy would be beneficial, as most papers do not use them.

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

## A    EXPERIMENTAL SETUP

### A.1    MACHINE LEARNING EFFICACY MODELS

For the Machine Learning Efficacy (MLE) score, we conducted a more in-depth experimentation with several other traditional classifiers. We selected the following diverse set of 7 machine learning models (results are shown in appendix C):

**RandomForest** was implemented using the `RandomForestClassifier` from the `scikit-learn` library.

**XGBoost** was implemented using the `XGBClassifier` from the `xgboost` library.

**LightGBM** was implemented using the `LGBMClassifier` from the `lightgbm` library.

**CatBoost** was implemented using the `CatBoostClassifier` from the `catboost` library.

**Logistic Regression** was implemented using the `LogisticRegression` class from the `scikit-learn` library.

**Support Vector Machines (SVM)** was implemented using the `SVC` class from the `scikit-learn` library.

**Multi-Layer Perceptrons (MLP)** was implemented using the `MLPClassifier` class from the `scikit-learn` library.

### A.2    FIDELITY METRICS

- **Wasserstein Distance (WD):** quantifies the cost of transforming the real distribution into the synthetic one and is particularly sensitive to shifts in tails and distribution spread. Lower WD indicates more accurate modeling of class-conditional distributions.
- **Jensen-Shannon Divergence (JSD):** measures the dissimilarity between probability distributions in a symmetric and bounded way. It captures how well the synthetic data approximates the global support and entropy of the real distribution.
- **Pairwise Correlation Error:** evaluates the structural consistency of synthetic data by computing the absolute difference between real and synthetic Pearson correlation matrices. This metric reflects how well inter-feature relationships are preserved.

### A.3    PRIVACY METRICS

Before computing privacy metrics (DCR and NNDR), we subsample 15% of real and synthetic data and apply z-score normalization. This ensures meaningful distance computations and consistency across datasets.

### A.4    PIPELINE

Figure 4 illustrates the experimental pipeline used in our study. The process begins with multiple tabular datasets, which are first preprocessed to ensure compatibility with all data generation models and downstream classifiers. This includes encoding categorical features, scaling numerical ones, and applying a fixed train/test split that preserves the original class imbalance ratio (IR).

The training set is then passed to a selected data generation methods. Each dataset is processed independently by each method to generate synthetic data that reflects the training distribution.

The synthetic data is then evaluated along 3 parallel axes: utility, fidelity and privacy analysis.

This dissected evaluation allows us to analyze each method's capacity to generate useful, faithful, and privacy-preserving synthetic data. The results are then aggregated and analyzed to draw conclusions about performance trade-offs and the effect of different techniques.

648
649
650
651
652
653
654
655
656
657
658
659
660
661
662
663
664
665
666
667
668
669
670
671
672
673
674
675
676
677
678
679
680
681
682
683
684
685
686
687
688
689
690
691
692
693
694
695
696
697
698
699
700
701

To ensure fair comparison, we fixed the random seed for all model initializations, training, and data splits. Each experiment was repeated with the same configuration across all methods.

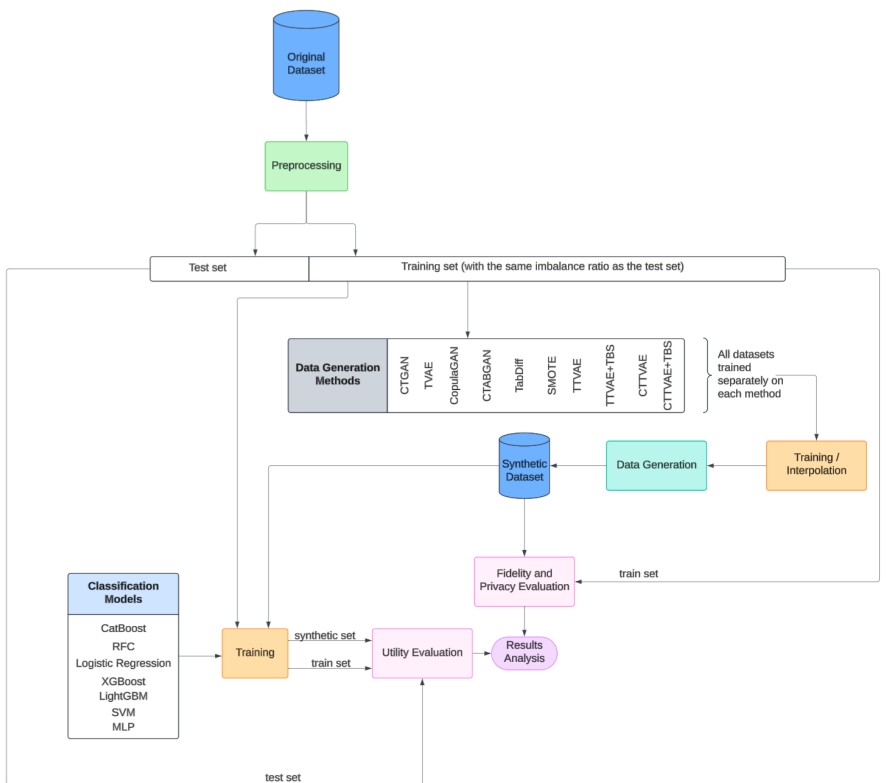

Figure 4: Pipeline

## A.5 IMPLEMENTATION OF BASELINE DATA GENERATION METHODS

To evaluate the performance of our proposed method, we implemented several baseline data generation methods commonly used for synthetic tabular data generation. We describe the implementation details for each method:

**SMOTE** was implemented using the `SMOTE` class from the `imblearn` library. A customized function was implemented to generate an entirely synthetic dataset.

**CTGAN** was implemented using the `CTGANSynthesizer` class from the `sdv` library.

**TVAE** was implemented using the `TVAESynthesizer` class from the `sdv` library.

**CopulaGAN** was implemented using the `CopulaGANSynthesizer` class from the `sdv` library.

**CTABGAN** was implemented using the code from its repository and adapted to our pipeline.

**TabDiff** was implemented using the code from its repository and adapted to our pipeline.

**TTVAE** was implemented using the code from its repository and adapted to our pipeline.

# B  SEMI-HARD TRIPLET MINING

Triplet mining is used to shape the latent space so that samples from the same class remain close while samples from different classes are pushed apart. In CTTVAE, we adopt a *semi-hard* triplet mining strategy, which selects triplets that are informative enough to improve class separation without being too difficult for the model to learn from. This improves the quality of the class-conditional latent structure, especially for minority classes, and directly supports more faithful conditional generation.

**Procedure.** We begin by computing all pairwise distances between the latent means $\{\mu_i\}_{i=1}^n$. For each sample $i$, we treat its latent vector $\mu_i$ as the *anchor* and partition the remaining samples into positives (same label) and negatives (different label). If the anchor has no valid positive or negative examples, it is skipped.

For every valid anchor, we select as the positive example the point of the same class that is *farthest* from the anchor. This choice encourages the encoder to reduce the intra-class spread by explicitly pulling difficult same-class samples closer together. We then look for *semi-hard negatives*, defined as negative samples whose distance to the anchor is greater than the anchor–positive distance but still within the margin $m$. These negatives are informative: they violate the desired class margin but are not so far away as to be irrelevant. If such candidates exist, one is chosen at random; otherwise, we fall back to the closest available negative.

Each anchor thus produces exactly one triplet consisting of the anchor, its hardest positive, and either a semi-hard or fallback closest negative. After processing all anchors, the triplet loss is computed as the average over all constructed triplets, encouraging the latent space to form compact, well-separated class clusters.

---

**Algorithm 1** Semi-hard triplet mining procedure for CTTVAE

---

Compute pairwise distances: $D \leftarrow \text{cdist}(\mu, \mu)$
**for** $i = 1$ to $n$ **do**
    $a \leftarrow \mu_i$                                                     ▷ anchor
    $\text{label}_a \leftarrow y_i$
    $\text{PosIndices} \leftarrow \{j \mid y_j = y_i, j \neq i\}$
    $\text{NegIndices} \leftarrow \{j \mid y_j \neq y_i\}$
    **if** $\text{PosIndices} = \emptyset$ or $\text{NegIndices} = \emptyset$ **then**
        **continue**
    **end if**
    $d_{ap} \leftarrow \min\{D[i][j] \mid j \in \text{PosIndices}\}$
    $\text{SemiHardMask} \leftarrow \{j \in \text{NegIndices} \mid d_{ap} < D[i][j] < d_{ap} + m\}$
    $\text{positive} \leftarrow \arg\max\{D[i][j] \mid j \in \text{PosIndices}\}$
    **if** $\text{SemiHardMask} \neq \emptyset$ **then**
        $\text{negative} \leftarrow$ random choice from SemiHardMask
    **else**
        $\text{negative} \leftarrow \arg\min\{D[i][j] \mid j \in \text{NegIndices}\}$
    **end if**
    Append triplet $(a, \text{positive}, \text{negative})$
**end for**
Compute average triplet loss over valid triplets

---

# C  UTILITY VS PRIVACY VS FIDELITY TRADEOFFS

To better understand how CTTVAE and the proposed training-by-sampling (TBS) strategy affect the minority class, we visualize the joint trade-offs between minority-class utility, privacy and fidelity for all generators. For each method $m$ and dataset $d$, we compute the ratio between the minority-class CatBoost MLE obtained on synthetic data and the corresponding score obtained on the real data, $\text{MLE}_{m,d}/\text{MLE}_{\text{Real},d}$, and aggregate these ratios across datasets using a geometric mean:

$$U_{\text{rel}}(m) = \exp\left(\frac{1}{|D|} \sum_{d \in D} \log \frac{\text{MLE}_{m,d}}{\text{MLE}_{\text{Real},d}}\right).$$

The $x$–axis in both plots reports this aggregate ratio: values around 1 indicate that the synthetic data supports a minority classifier that is as strong as the one trained on real data, values $< 1$ indicate utility loss, and values $> 1$ indicate a net gain in minority-class utility.

In Figure 5 (a), the $y$–axis shows the same ratio definition but applied to the minority-class privacy score (NNDR, higher is better). In Figure 5 (b), the $y$–axis reports the ratio of minority-class Wasserstein distance (WD), where lower is better: points closer to $(1, 0)$ correspond to synthetic data that matches the real distribution while preserving high minority utility. The point labelled *Optimal* is an unattainable reference corresponding to perfect privacy (NNDR ratio $= 1$) and zero fidelity error (WD ratio $= 0$) while matching or exceeding real-data utility (utility ratio $= 1$).

Across both views, the three CTTVAE-based variants occupy a favorable region of the trade-off. CTTVAE+TBS lies at the extreme right of the plot (utility ratio $\approx 1$), indicating that it essentially recovers real-data minority performance while preserving competitive privacy and low fidelity error. CTTVAE (without TBS) already improves over classical baselines such as CTGAN, TVAE, CopulaGAN and CTABGAN, but TBS consistently shifts the CTTVAE point further towards the desired region (higher utility with only a mild change in NNDR and a modest increase in WD). A similar behaviour is observed when moving from TTVAE to TTVAE+TBS: the TBS variant achieves clearly higher minority utility for comparable privacy and only a slight degradation in WD. TabDiff, which is a strong diffusion-based baseline, attains solid privacy and fidelity but remains noticeably to the left of CTTVAE+TBS on the utility axis, illustrating that its minority samples are less useful for downstream classification than those generated by our method.

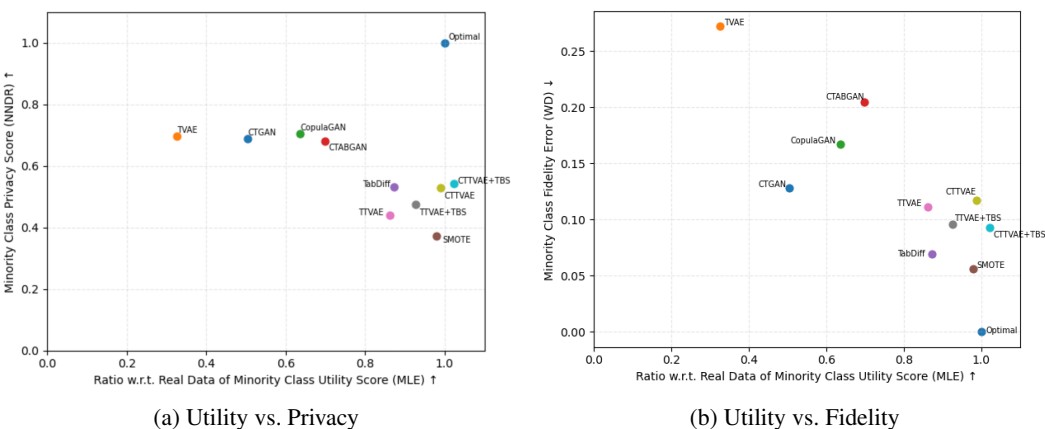

(a) Utility vs. Privacy             (b) Utility vs. Fidelity

Figure 5: Minority-class trade-offs between utility and (a) privacy and (b) fidelity. The $x$–axis shows the geometric mean ratio of minority CatBoost MLE on synthetic vs. real data (higher is better). The $y$–axis reports the corresponding ratio for minority NNDR (higher is better) or WD (lower is better). The point labelled *Optimal* is a conceptual reference corresponding to matching real-data utility with perfect privacy or zero fidelity error. CTTVAE, TTVAE+TBS and CTTVAE+TBS all lie in the desirable high-utility region, with CTTVAE+TBS achieving the best minority utility while retaining competitive privacy and fidelity compared to both classical GAN/VAE baselines and the diffusion-based TabDiff.

## D  DATASET PREPROCESSING DETAILS

All datasets used in this study are publicly available, and the corresponding preprocessing code is provided in the official repository, with dedicated notebooks for each dataset. Preprocessing involved only minimal cleaning: removing rows with missing values or duplicates, digitizing target columns, and dropping irrelevant features such as IDs.

# E ADDITIONAL RESULTS

Table 6 reports the mean and standard deviation of the MLE scores over three generations, separated by majority and minority classes. As expected, majority-class performance remains highly stable across all methods, with very low variance, while minority-class results show larger fluctuations reflecting the higher sensitivity to imbalance. This confirms that our improvements primarily benefit the minority class without degrading performance on the majority.

Table 6: Average MLE and standard deviation computed with CatBoost across datasets for each class group (Maj = Majority, Min = Minority). **Bold** represents the best results on each dataset and underlined represents the second best results for minority samples only on each dataset. Its performance on the majority class remains stable for all the considered methods. "Real" represents the scores of CatBoost trained on the original dataset. Higher means better.

(a) Majority Class Only

| Method | CH | AD | DE | CR | MA | VE |
|---|---|---|---|---|---|---|
| Real | 0.923±0.001 | 0.918±0.002 | 0.890±0.003 | 0.999±0.001 | 0.994±0.004 | 0.970±0.002 |
| CTGAN | 0.887±0.004 | 0.906±0.007 | 0.870±0.003 | 0.993±0.002 | 0.949±0.004 | 0.970±0.001 |
| TVAE | 0.889±0.001 | 0.881±0.005 | 0.885±0.002 | 0.998±0.001 | 0.982±0.001 | 0.970±0.001 |
| CopulaGAN | 0.856±0.001 | 0.894±0.004 | 0.883±0.003 | 0.981±0.002 | 0.940±0.003 | 0.970±0.001 |
| CTABGAN | 0.894±0.002 | 0.896±0.004 | 0.868±0.004 | 0.998±0.001 | 0.983±0.002 | 0.962±0.002 |
| TabDiff | 0.920±0.001 | 0.912±0.001 | 0.891±0.001 | 0.999±0.001 | 0.989±0.002 | 0.970±0.001 |
| SMOTE | 0.917±0.001 | 0.908±0.001 | 0.882±0.001 | 0.999±0.001 | 0.990±0.001 | 0.969±0.001 |
| TTVAE | 0.919±0.001 | 0.910±0.003 | 0.890±0.002 | 0.999±0.001 | 0.989±0.002 | 0.968±0.001 |
| TTVAE+TBS | 0.924±0.001 | 0.911±0.001 | 0.881±0.001 | 0.999±0.001 | 0.990±0.001 | 0.970±0.001 |
| CTTVAE | 0.921±0.001 | 0.910±0.001 | 0.877±0.001 | 0.999±0.001 | 0.989±0.001 | 0.968±0.001 |
| CTTVAE+TBS | 0.920±0.001 | 0.910±0.002 | 0.882±0.002 | 0.999±0.001 | 0.991±0.001 | 0.967±0.001 |

(b) Minority Class Only

| Method | CH | AD | DE | CR | MA | VE |
|---|---|---|---|---|---|---|
| Real | 0.607±0.001 | 0.728±0.002 | 0.468±0.003 | 0.893±0.001 | 0.790±0.004 | 0.112±0.002 |
| CTGAN | 0.559±0.042 | 0.677±0.001 | 0.459±0.020 | 0.428±0.161 | 0.327±0.010 | 0.011±0.010 |
| TVAE | 0.502±0.015 | 0.609±0.003 | 0.397±0.006 | 0.838±0.020 | 0.189±0.006 | 0.001±0.001 |
| CopulaGAN | 0.560±0.010 | 0.569±0.004 | 0.474±0.038 | 0.450±0.190 | 0.302±0.023 | 0.053±0.060 |
| CTABGAN | 0.575±0.020 | 0.612±0.002 | 0.466±0.039 | 0.498±0.172 | 0.327±0.006 | 0.071±0.012 |
| TabDiff | 0.574±0.010 | 0.679±0.002 | 0.478±0.001 | 0.869±0.012. | 0.628±0.083 | 0.071±0.006 |
| SMOTE | 0.608±0.014 | 0.694±0.001 | 0.501±0.001 | **0.891±0.001** | **0.678±0.035** | 0.113±0.018 |
| TTVAE | 0.607±0.004 | 0.689±0.001 | 0.463±0.004 | 0.857±0.012 | 0.560±0.017 | 0.072±0.002 |
| TTVAE+TBS | 0.606±0.003 | 0.703±0.002 | 0.498±0.003 | 0.867±0.010 | 0.667±0.024 | 0.084±0.009 |
| CTTVAE | 0.627±0.005 | 0.669±0.004 | 0.495±0.005 | 0.881±0.003 | 0.637±0.039 | 0.131±0.011 |
| CTTVAE+TBS | **0.628±0.006** | **0.703±0.002** | **0.512±0.009** | 0.881±0.004 | **0.684±0.045** | **0.137±0.016** |

In Table 7, we compute the average F1 score across 7 classifiers for each method and dataset. A higher MLE indicates better alignment with real-data performance, suggesting greater practical utility of the synthetic data.

The per-class Wasserstein Distance results across datasets are presented in Table 8, separated into moderately and highly imbalanced datasets. The per-class Jensen-Shannon Divergence scores across datasets are shown in Table 9. The CR dataset is omitted from this table due to its lack of categorical features.

Table 7: The values of the average MLE and standard deviation for each method and each dataset averaged over all classifiers. Each classifier has been tuned and then trained 10 times (training not seeded) with the best set of hyperparameters on the same generated data. **Bold** represents the best results on each dataset and underlined represents the second best results on each dataset. "Real" represents the scores of the models trained on the original dataset. Higher means better.

| Method | CH | AD | DE | CR | MA | VE |
|---|---|---|---|---|---|---|
| Real | 0.732±0.001 | 0.811±0.001 | 0.670±0.001 | 0.945±0.001 | 0.826±0.004 | 0.530±0.002 |
| CTGAN | 0.697±0.001 | 0.785±0.001 | 0.676±0.001 | 0.815±0.004 | 0.633±0.002 | 0.489±0.002 |
| TVAE | 0.698±0.001 | 0.747±0.001 | 0.646±0.001 | 0.858±0.004 | 0.580±0.003 | 0.485±0.000 |
| CopulaGAN | 0.706±0.001 | 0.725±0.001 | 0.646±0.002 | 0.597±0.009 | 0.612±0.002 | 0.487±0.002 |
| CTABGAN | 0.711±0.002 | 0.752±0.001 | 0.667±0.001 | 0.836±0.004 | 0.606±0.006 | 0.518±0.002 |
| TabDiff | 0.706±0.005 | 0.788±0.001 | 0.668±0.003 | 0.913±0.004 | 0.717±0.038 | 0.508±0.012 |
| SMOTE | 0.735±0.001 | 0.797±0.001 | **0.685±0.001** | **0.943±0.001** | **0.799±0.003** | **0.537±0.002** |
| TTVAE | 0.735±0.001 | 0.797±0.001 | 0.656±0.002 | 0.925±0.001 | 0.710±0.006 | 0.507±0.002 |
| TTVAE+TBS | 0.725±0.002 | 0.798±0.001 | 0.680±0.001 | 0.916±0.001 | 0.746±0.003 | 0.510±0.002 |
| CTTVAE | **0.744±0.001** | 0.792±0.001 | 0.676±0.003 | 0.930±0.003 | 0.772±0.005 | 0.529±0.003 |
| CTTVAE+TBS | **0.744±0.001** | **0.801±0.001** | 0.683±0.001 | 0.927±0.004 | 0.774±0.004 | 0.531±0.002 |

Table 8: Wasserstein Distance per class average and standard deviation over three generations across datasets. **Bold** represents the best results and underline represents the second best. Lower is better.

(a) Moderately imbalanced datasets

| | CH | | AD | | DE | |
|---|---|---|---|---|---|---|
| | Maj. | Min. | Maj. | Min. | Maj. | Min. |
| CTGAN | 0.109±0.033 | 0.125±0.035 | 0.131±0.062 | 0.110±0.051 | 0.074±0.037 | 0.082±0.035 |
| TVAE | 0.242±0.091 | 0.268±0.084 | 0.184±0.094 | 0.204±0.093 | 0.107±0.056 | 0.331±0.168 |
| CopulaGAN | 0.142±0.070 | 0.148±0.086 | 0.136±0.043 | 0.177±0.042 | 0.103±0.052 | 0.162±0.086 |
| CTABGAN | 0.187±0.075 | 0.182±0.067 | 0.312±0.136 | 0.337±0.151 | 0.200±0.099 | 0.243±0.141 |
| TabDiff | 0.043±0.014 | 0.053±0.003 | **0.032±0.004** | **0.037±0.006** | **0.026±0.004** | **0.044±0.006** |
| SMOTE | 0.037±0.010 | **0.040±0.011** | 0.051±0.021 | 0.065±0.026 | 0.042±0.021 | 0.063±0.029 |
| TTVAE | **0.032±0.013** | 0.064±0.024 | 0.067±0.026 | 0.097±0.037 | 0.066±0.029 | 0.103±0.046 |
| TTVAE+TBS | 0.041±0.022 | 0.060±0.031 | 0.074±0.057 | 0.091±0.064 | 0.101±0.078 | 0.108±0.080 |
| CTTVAE | 0.044±0.017 | 0.046±0.021 | 0.063±0.032 | 0.096±0.050 | 0.085±0.042 | 0.106±0.044 |
| CTTVAE+TBS | 0.039±0.024 | 0.056±0.033 | 0.052±0.029 | 0.072±0.042 | 0.089±0.049 | 0.116±0.060 |

(b) Highly imbalanced datasets

| | CR | | MA | | VE | |
|---|---|---|---|---|---|---|
| | Maj. | Min. | Maj. | Min. | Maj. | Min. |
| CTGAN | 0.136±0.070 | 0.186±0.073 | 0.080±0.018 | 0.150±0.048 | 0.090±0.033 | 0.106±0.025 |
| TVAE | 0.101±0.048 | 0.260±0.087 | 0.125±0.043 | 0.210±0.072 | 0.053±0.024 | 0.362±0.165 |
| CopulaGAN | 0.181±0.065 | 0.256±0.085 | 0.105±0.030 | 0.176±0.046 | 0.071±0.010 | 0.083±0.016 |
| CTABGAN | 0.128±0.097 | 0.231±0.108 | 0.035±0.010 | 0.133±0.049 | 0.096±0.035 | 0.105±0.039 |
| TabDiff | 0.030±0.006 | 0.148±0.045 | 0.017±0.002 | 0.081±0.002 | 0.032±0.009 | 0.051±0.002 |
| SMOTE | **0.019±0.009** | **0.078±0.033** | 0.019±0.004 | **0.048±0.015** | **0.020±0.009** | **0.040±0.015** |
| TTVAE | 0.143±0.090 | 0.249±0.081 | **0.013±0.005** | 0.099±0.031 | 0.023±0.010 | 0.052±0.012 |
| TTVAE+TBS | 0.186±0.135 | 0.200±0.082 | 0.017±0.011 | 0.072±0.040 | 0.029±0.030 | 0.047±0.024 |
| CTTVAE | 0.137±0.069 | 0.210±0.094 | **0.013±0.006** | 0.158±0.027 | 0.065±0.015 | 0.078±0.021 |
| CTTVAE+TBS | 0.133±0.098 | 0.147±0.099 | 0.015±0.006 | 0.091±0.025 | 0.065±0.027 | 0.074±0.024 |

Table 10 reports the pairwise correlation error rates across datasets. SMOTE achieves the lowest correlation errors in most cases, particularly on CR and DE. CTTVAE and its TBS variant also perform well, with notably low errors and comparable with the baseline interpolation methods. In contrast, models like CTGAN and CopulaGAN show higher deviation from the real data's correlation structure.

Table 9: Jensen-Shannon Divergence per class average and standard deviation over three generations across datasets. **Bold** represents the best results and underline represents the second best. Lower scores are better. CR dataset is omitted since it does not contain categorical features.

(a) Moderately imbalanced datasets

| | CH | | AD | | DE | |
|---|---|---|---|---|---|---|
| | Maj. | Min. | Maj. | Min. | Maj. | Min. |
| CTGAN | 0.024±0.014 | 0.025±0.014 | 0.101±0.052 | 0.104±0.042 | 0.116±0.053 | 0.086±0.039 |
| TVAE | 0.224±0.095 | 0.232±0.098 | 0.089±0.047 | 0.097±0.050 | 0.157±0.067 | 0.172±0.073 |
| CopulaGAN | 0.028±0.015 | 0.033±0.018 | 0.104±0.053 | 0.107±0.043 | 0.103±0.044 | 0.094±0.039 |
| CTABGAN | 0.052±0.028 | 0.057±0.032 | 0.143±0.069 | 0.140±0.068 | 0.057±0.030 | 0.070±0.034 |
| TabDiff | 0.015±0.008 | 0.010±0.004 | 0.029±0.004 | 0.049±0.001 | 0.042±0.006 | 0.046±0.006 |
| SMOTE | **0.004±0.002** | **0.012±0.008** | **0.009±0.005** | **0.018±0.010** | **0.003±0.002** | **0.008±0.004** |
| TTVAE | 0.012±0.006 | 0.019±0.009 | 0.039±0.023 | 0.051±0.027 | 0.040±0.013 | 0.035±0.011 |
| TTVAE+TBS | 0.016±0.011 | 0.022±0.012 | 0.068±0.048 | 0.066±0.042 | 0.064±0.037 | 0.083±0.046 |
| CTTVAE | 0.009±0.008 | 0.018±0.013 | 0.041±0.022 | 0.056±0.033 | 0.073±0.032 | 0.067±0.029 |
| CTTVAE+TBS | 0.009±0.004 | 0.016±0.009 | 0.045±0.013 | 0.058±0.021 | 0.067±0.033 | 0.060±0.027 |

(b) Highly imbalanced datasets

| | MA | | VE | |
|---|---|---|---|---|
| | Maj. | Min. | Maj. | Min. |
| CTGAN | 0.066±0.032 | 0.092±0.045 | 0.115±0.054 | 0.151±0.071 |
| TVAE | 0.114±0.048 | 0.091±0.040 | 0.118±0.051 | 0.296±0.115 |
| CopulaGAN | 0.100±0.043 | 0.123±0.052 | 0.124±0.050 | 0.144±0.069 |
| CTABGAN | 0.030±0.016 | **0.008±0.004** | 0.098±0.044 | 0.115±0.057 |
| TabDiff | 0.020±0.001 | 0.014±0.010 | 0.027±0.006 | 0.050±0.003 |
| SMOTE | **0.010±0.006** | 0.011±0.006 | **0.020±0.010** | 0.044±0.024 |
| TTVAE | 0.017±0.009 | 0.060±0.029 | 0.031±0.013 | 0.055±0.027 |
| TTVAE+TBS | 0.023±0.011 | 0.035±0.020 | 0.031±0.034 | **0.042±0.039** |
| CTTVAE | 0.025±0.012 | 0.022±0.021 | 0.053±0.016 | 0.070±0.027 |
| CTTVAE+TBS | 0.018±0.008 | 0.050±0.007 | 0.038±0.012 | 0.060±0.026 |

Table 10: Pair-wise correlation error rate (%) averaged over three generations with standard deviation for each method across datasets. **Bold** represents the best results and underline represents the second best on each dataset. Lower scores means better.

| Method | CH | AD | DE | CR | MA | VE |
|---|---|---|---|---|---|---|
| CTGAN | 2.89±0.13 | 2.40±0.21 | 3.39±1.86 | 25.71±7.51 | 29.95±6.24 | 4.55±0.31 |
| TVAE | 8.97±1.85 | 4.86±2.07 | 5.21±1.07 | 11.29±2.05 | 3.93±1.32 | 4.48±0.37 |
| CopulaGAN | 3.15±0.83 | 3.32±0.65 | 5.85±0.97 | 32.32±18.26 | 27.90±5.14 | 4.37±0.55 |
| CTABGAN | 3.94±0.39 | 6.89±3.17 | 8.54±1.13 | 6.79±3.50 | 3.19±1.57 | 7.99±1.26 |
| TabDiff | 1.99±0.11 | 1.87±0.07 | **1.22±0.08** | 3.12±0.62 | 2.83±0.25 | 1.57±0.17 |
| SMOTE | **1.05±0.10** | 1.18±0.08 | 1.78±0.07 | **1.82±0.08** | 1.32±0.011 | **1.42±0.20** |
| TTVAE | 1.12±0.07 | **1.05±0.09** | 2.31±0.37 | 5.20±0.49 | 1.39±0.16 | 1.76±0.20 |
| TTVAE+TBS | 1.43±0.14 | 1.14±0.15 | 3.22±0.45 | 5.31±0.48 | 1.43±0.22 | 1.52±0.18 |
| CTTVAE | 1.08±0.09 | 1.37±0.20 | 2.31±0.28 | 6.31±0.60 | 1.14±0.07 | 1.67±0.17 |
| CTTVAE+TBS | 1.13±0.06 | 1.40±0.15 | 2.29±0.23 | 5.23±0.53 | **0.95±0.06** | 1.63±0.15 |

Table 11 summarizes the results. Tables 12 13 report per-class privacy scores. Higher values indicate greater dissimilarity between synthetic and real records, which typically suggests better privacy preservation. However, high DCR and NNDR can sometimes reflect low data utility and fidelity if the synthetic samples drift too far from the true data distribution. For instance, COPULAGAN and CTABGAN achieve consistently among the highest scores but often performs poorly in terms of utility. This does not imply that the synthetic data is of high quality. On the contrary, it instead signals poor alignment with the original data.

Among the generative models, TTVAE and CTTVAE variants tend to strike a more balanced profile, achieving moderate scores without overstepping into unrealistic territory given their high utility scores. In highly imbalanced settings, TTVAE-based model achieve strong comparable privacy scores w.r.t. other methods, suggesting that these methods and training strategies are more suitable for these types of datasets. Still, it is crucial to interpret DCR and NNDR jointly with fidelity and utility metrics as it does not paint the full picture.

Table 11: Overall average and standard deviation Distance to Closest Record (DCR) and Nearest Neighbour Distance Ratio (NNDR) across all datasets. **Bold** represents the best results and underline represents the second best. Higher values indicate better privacy.

| Method | DCR ↑ | | NNDR ↑ | |
|---|---|---|---|---|
| | Maj. | Min. | Maj. | Min. |
| CTGAN | 2.379±3.216 | 2.587±3.641 | 0.665±0.191 | 0.690±0.174 |
| TVAE | 2.129±2.553 | 1.933±1.415 | **0.759±0.125** | 0.697±0.165 |
| CopulaGAN | **2.386±3.265** | 2.596±3.474 | 0.697±0.165 | **0.705±0.183** |
| CTABGAN | 2.230±2.878 | **2.692±3.059** | 0.707±0.129 | 0.682±0.184 |
| TabDiff | 1.283±2.123 | 1.637±2.187 | 0.498±0.134 | 0.533±0.136 |
| SMOTE | 0.380±0.117 | 0.864±0.557 | 0.282±0.132 | 0.372±0.161 |
| TTVAE | 0.699±0.716 | 1.383±1.352 | 0.368±0.247 | 0.440±0.183 |
| TTVAE+TBS | 0.774±0.928 | 1.262±1.055 | 0.393±0.287 | 0.476±0.202 |
| CTTVAE | 1.587±2.443 | 1.834±2.176 | 0.529±0.190 | 0.528±0.163 |
| CTTVAE+TBS | 1.587±2.254 | 1.528±1.699 | 0.534±0.187 | 0.543±0.140 |

Table 12: Average and standard deviation per-class privacy scores (DCR) over three generations across moderately and highly imbalanced datasets. **Bold** represents the best results and underline represents the second best on each dataset. Higher scores means better.

(a) DCR – Moderately imbalanced datasets

| | CH | | AD | | DE | |
|---|---|---|---|---|---|---|
| | Maj. | Min. | Maj. | Min. | Maj. | Min. |
| CTGAN | 0.692±0.097 | 0.993±0.100 | 1.000±0.135 | 0.912±0.112 | 0.712±0.145 | 0.876±0.178 |
| TVAE | **1.501±0.215** | **1.664±0.248** | 0.774±0.188 | 0.890±0.225 | **1.009±0.265** | **1.778±0.310** |
| CopulaGAN | 0.750±0.165 | 0.802±0.175 | 0.970±0.203 | 1.055±0.240 | 0.775±0.195 | 1.196±0.245 |
| CTABGAN | 0.980±0.190 | 1.079±0.220 | **1.474±0.260** | **1.745±0.300** | 0.810±0.215 | 1.271±0.265 |
| TabDiff | 0.609±0.012 | 0.676±0.042 | 0.362±0.020 | 0.470±0.038 | 0.313±0.001 | 0.555±0.008 |
| SMOTE | 0.356±0.021 | 0.517±0.045 | 0.368±0.030 | 0.482±0.042 | 0.302±0.018 | 0.497±0.024 |
| TTVAE | 0.179±0.015 | 0.344±0.023 | 0.390±0.025 | 0.483±0.019 | 0.359±0.043 | 0.656±0.065 |
| TTVAE+TBS | 0.194±0.014 | 0.565±0.131 | 0.469±0.043 | 0.556±0.013 | 0.542±0.076 | 0.841±0.070 |
| CTTVAE | 0.375±0.22 | 0.482±0.015 | 0.423±0.024 | 0.604±0.030 | 0.509±0.010 | 0.787±0.040 |
| CTTVAE+TBS | 0.380±0.019 | 0.480±0.013 | 0.468±0.037 | 0.628±0.083 | 0.650±0.031 | 0.589±0.044 |

(b) DCR – Highly imbalanced datasets

| | CR | | MA | | VE | |
|---|---|---|---|---|---|---|
| | Maj. | Min. | Maj. | Min. | Maj. | Min. |
| CTGAN | 2.613±0.180 | 2.320±0.200 | 0.505±0.021 | 0.512±0.051 | 8.753±0.400 | **9.910±0.460** |
| TVAE | 1.799±0.200 | **4.481±0.280** | 0.447±0.040 | 0.849±0.075 | 7.244±0.340 | 0.750±0.100 |
| CopulaGAN | 2.189±0.193 | 2.042±0.250 | **0.683±0.064** | **0.850±0.076** | **8.950±0.391** | 9.628±0.422 |
| CTABGAN | 1.789±0.220 | 2.799±0.290 | 0.316±0.036 | 0.521±0.046 | 8.011±0.300 | 8.736±0.350 |
| TabDiff | 0.526±0.048 | 1.842±0.078 | 0.277±0.001 | 0.322±0.011 | 5.609±0.266 | 5.959±0.331 |
| SMOTE | 0.454±0.037 | 1.458±0.067 | 0.233±0.012 | 0.534±0.062 | 0.565±0.212 | 1.693±0.218 |
| TTVAE | 1.955±0.183 | 3.160±0.427 | 0.154±0.003 | 0.568±0.039 | 1.158±0.317 | 3.084±1.383 |
| TTVAE+TBS | **2.622±0.209** | 3.044±0.456 | 0.143±0.023 | 0.502±0.041 | 0.676±0.014 | 2.066±0.124 |
| CTTVAE | 1.488±0.038 | 2.569±0.084 | 0.240±0.024 | 0.590±0.032 | 6.489±0.543 | 5.974±0.630 |
| CTTVAE+TBS | 1.753±0.423 | 2.281±0.096 | 0.219±0.014 | 0.497±0.025 | 6.051±0.282 | 4.691±0.734 |

Table 13: Average NNDR and standard deviation per-class privacy scores over three generations across moderately and highly imbalanced datasets. **Bold** represents the best results and underline represents the second best on each dataset. Higher scores means better.

(a) NNDR – Moderately imbalanced datasets

| | CH | | AD | | DE | |
|---|---|---|---|---|---|---|
| | Maj. | Min. | Maj. | Min. | Maj. | Min. |
| CTGAN | 0.496±0.032 | 0.541±0.036 | 0.413±0.037 | 0.469±0.039 | 0.684±0.051 | 0.665±0.047 |
| TVAE | **0.842±0.102** | **0.848±0.098** | 0.524±0.045 | 0.535±0.053 | **0.816±0.099** | **0.859±0.092** |
| CopulaGAN | 0.540±0.029 | 0.511±0.032 | 0.490±0.030 | 0.479±0.033 | 0.709±0.046 | 0.699±0.048 |
| CTABGAN | 0.612±0.035 | 0.588±0.034 | **0.606±0.050** | **0.748±0.063** | 0.761±0.038 | 0.748±0.038 |
| TabDiff | 0.485±0.005 | 0.489±0.018 | 0.320±0.012 | 0.371±0.033 | 0.511±0.004 | 0.486±0.018 |
| SMOTE | 0.276±0.020 | 0.307±0.023 | 0.203±0.017 | 0.263±0.020 | 0.341±0.023 | 0.347±0.024 |
| TTVAE | 0.136±0.026 | 0.240±0.030 | 0.276±0.021 | 0.345±0.025 | 0.502±0.027 | 0.538±0.028 |
| TTVAE+TBS | 0.169±0.021 | 0.336±0.027 | 0.349±0.024 | 0.447±0.040 | 0.636±0.034 | 0.601±0.038 |
| CTTVAE | 0.342±0.025 | 0.327±0.026 | 0.340±0.022 | 0.402±0.027 | 0.618±0.031 | 0.582±0.029 |
| CTTVAE+TBS | 0.336±0.023 | 0.344±0.028 | 0.354±0.023 | 0.420±0.028 | 0.629±0.031 | 0.621±0.031 |

(b) NNDR – Highly imbalanced datasets

| | CR | | MA | | VE | |
|---|---|---|---|---|---|---|
| | Maj. | Min. | Maj. | Min. | Maj. | Min. |
| CTGAN | 0.877±0.072 | 0.893±0.075 | 0.644±0.021 | 0.685±0.052 | **0.878±0.040** | 0.888±0.046 |
| TVAE | 0.807±0.058 | 0.834±0.066 | **0.719±0.035** | 0.567±0.034 | 0.851±0.040 | 0.540±0.042 |
| CopulaGAN | **0.886±0.079** | **0.912±0.068** | 0.679±0.064 | **0.738±0.075** | **0.878±0.055** | **0.892±0.049** |
| CTABGAN | 0.834±0.103 | 0.822±0.121 | 0.565±0.038 | 0.351±0.036 | 0.866±0.077 | 0.835±0.074 |
| TabDiff | 0.538±0.019 | 0.667±0.136 | 0.416±0.008 | 0.456±0.063 | 0.721±0.025 | 0.730±0.017 |
| SMOTE | 0.353±0.020 | 0.537±0.029 | 0.448±0.021 | 0.594±0.028 | 0.070±0.012 | 0.183±0.018 |
| TTVAE | 0.798±0.039 | 0.738±0.035 | 0.319±0.024 | 0.477±0.030 | 0.175±0.020 | 0.302±0.024 |
| TTVAE+TBS | 0.835±0.041 | 0.766±0.036 | 0.282±0.023 | 0.514±0.032 | 0.089±0.015 | 0.189±0.020 |
| CTTVAE | 0.756±0.037 | 0.790±0.038 | 0.404±0.026 | 0.481±0.031 | 0.715±0.040 | 0.584±0.036 |
| CTTVAE+TBS | 0.784±0.038 | 0.730±0.036 | 0.428±0.027 | 0.570±0.034 | 0.674±0.038 | 0.572±0.035 |

## F ADDITIONAL VISUALIZATIONS

The heatmaps in Figure 6 provide a complementary view of fidelity by visualizing how well the correlation structure of the real data is preserved across models in the ablation study. As shown, CTTVAE+TBS maintains lighter patterns compared to alternatives, indicating lower deviation from the real correlation structure. This visualization confirms the quantitative fidelity results, where our proposed method remains competitive with the strongest baselines while offering superior utility for minority classes.

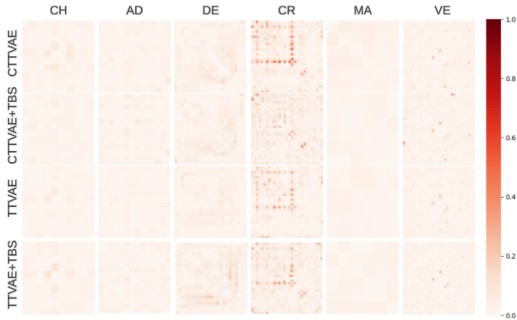

Figure 6: The absolute difference between correlation matrices computed on real and synthetic datasets for the ablation study. More intense red color indicates higher difference.

PCA projections in Figure 7 reveal that CTTVAE yields clearer class boundaries and tighter clusters than TTVAE, confirming the role of triplet loss in enabling coherent, class-aware generation under imbalance. Furthermore, we see that the clusters keep a non spherical shape, allowing for outliers to remain as such (as opposed to how contrastive losses separate the space). Maintaining outliers is important, especially in imbalanced settings since often those are the most important instances.

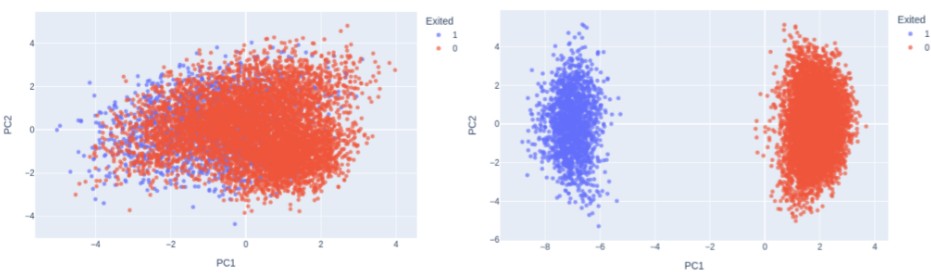

Figure 7: Latent space encoded by TTVAE (left) and CTTVAE (right) for the CH dataset, projected on a 2D space using PCA for visualization purposes.

## G RUNTIME

We report the training and sampling time of CTTVAE and TTVAE for the Churn Modeling (CH) dataset for comparison (Table 14). Both models have been trained on A100 GPU. Although CTTVAE training is slower due to triplet mining, the overhead remains modest relative to modern GPU capabilities, and the resulting gains in minority-class utility outweigh this cost. Efficient mining or adaptive margins can further reduce runtime.

Table 14: Training and sampling time for CTTVAE and TTVAE for the CH dataset.

(a) **Training time**

| Model | batch_size | epochs | train_steps | training time |
|---|---|---|---|---|
| TTVAE | 128 | 125 | 7,812 | 381s |
| CTTVAE | 128 | 125 | 7,812 | 803s |

(b) **Sampling time**

| Model | number to sample | sample_time |
|---|---|---|
| TTVAE | 8k | 0.29s |
| CTTVAE | 8k | 0.33s |

## H HYPERPARAMETER SEARCH SPACES

We performed hyperparameter optimization using Optuna library for both the downstream MLE classifiers (Table 15) and the generative models (Table 16). For each generative model, we conducted 25 trials to identify the best-performing configuration based on utility scores. Due to computational constraints, hyperparameter tuning for CTTVAE and TTVAE was divided into two stages: we first selected the best model configuration and then conducted a focused search on the L2 regularization scale for both models and the triplet loss factor for CTTVAE. For experiments involving TBS, we did not run a full 25-trial search; instead, we evaluated different values of the sampling hyperparameter $\lambda$ using the previously selected best configuration for each model.

Table 15: Hyperparameter search space for classifier models used for MLE

| Model | Search Space |
|---|---|
| RandomForest | num estimators: Int[50, 300]
max depth: Int[3, 20]
min samples_split: Int[2, 10]
min samples_leaf: Int[1, 10] |
| XGBoost | n_estimators: Int[50, 300]
max_depth: Int[3, 20]
learning rate: Float[0.01, 0.3] |
| LightGBM | num estimators: Int[50, 300]
num leaves: Int[20, 100]
learning rate: Float[0.01, 0.3] |
| CatBoost | iterations: Int[50, 300]
depth: Int[3, 10]
learning rate: Float[0.01, 0.3] |
| LogisticRegression | C: Float[0.01, 10.0]
penalty: {l1, l2}
solver: {liblinear, saga} |
| SVM | C: Float[0.01, 10.0]
kernel: {linear, rbf} |
| MLP | hidden layer: {(100,), (50,50), (100,50)}
activation: {relu, tanh}
alpha: Float[1e-5, 1e-1]
max iter: 500 (fixed) |
| Number of tuning trials | 30 |

Table 16: Hyperparameter search space for deep generative models.

| Model | Search Space |
|---|---|
| CTGAN / CopulaGAN | pac: {1, 5, 10}
batch_size: {64, 128, 256, 500}
epochs: {50, 100, 150} |
| TVAE | batch_size: {64, 128, 256, 512}
epochs: {10, 50, 100, 150} |
| CTABGAN | batch_size: {64, 128, 256}
epochs: {150, 200, 250}
class_dim: {128, 256}
l2scale: Float[1e-6, 1e-3]
learning rate: Float[1e-4, 1e-2]
num_channels: {32, 64, 128}
random_dim: {64, 100, 128} |
| TabDiff | steps: {1000 to 8000}
num_timesteps: {25, 50, 100} |
| TTVAE | batch_size: {16, 32, 64}
epochs: {10, 50, 100, 150}
latent_dim: {16, 32, 64}
embedding_dim: {64, 128, 256}
nhead: derived from (64,4), (128,4/8), (256,8)
dim_feedforward: {512, 1024, 2048}
dropout: Float[0.0, 0.3]
l2scale: {1e-5, 1e-4, 1e-3} |
| CTTVAE | batch_size: {16, 32, 64}
epochs: {10, 50, 100, 150}
latent_dim: {16, 32, 64}
embedding_dim: {64, 128, 256}
nhead: derived from (64,4), (128,4/8), (256,8)
dim_feedforward: {512, 1024, 2048}
dropout: Float[0.0, 0.3]
triplet_margin: Float[0.1, 1.0]
l2scale: {1e-5, 1e-4, 1e-3}
triplet_factor: {0.5, 1, 2, 5} |
| TBS | $\lambda$: {0.3, 0.5,0.7, 0.9} |
| Number of tuning trials | 25 |

