# OpenReview forum: "Latent Space Structuring for Conditional Tabular Data Generation on Imbalanced Datasets"
_ICLR.cc/2026/Conference — Submitted to ICLR 2026_

### Official Review · Reviewer_ZBoB · 2025-10-27

**Soundness:** 2
**Presentation:** 3
**Contribution:** 2
**Rating:** 2
**Confidence:** 5

**Summary:**

The paper proposes CTTVAE+TBS which is a transformer-based tabular VAE that can perform synthetic data generation for imbalanced tabular data. It has 2 key components: CTTVAE (class-aware triplet loss) and TBS (training by sampling). The authors find on the 6 real world datasets that was tested, the proposed method is able to yield a higher F1 score (high utility) while maintaining fidelity (measured by Wasserstein and JSD) and also preserve privacy (measured by DCR and NNDR).

**Strengths:**

1. The proposed architecture combined with upsampling has not been done before
2. The results look great in terms of all 3 aspects of tabular synthetic data generated
3. The paper is well written and the proposed concepts are presented clearly

**Weaknesses:**

1. Only 6 datasets are used. It is lot fewer than other past works, which makes the claim less strong.
2. Diffusion method such as TabDDPM and tabular foundation model based methods such as TabPFGen and TabEBM are not present in the results

**Questions:**

1. It would be great to show an intuitive 2D dataset and show how the proposed method is able to outperform other baseline methods

---

> ### Author Response · Authors · 2025-11-25
> **Response to Reviewer ZBoB**
>
> We thank the reviewer for the comments and for highlighting the strengths of our method. We respectfully address the concerns below.
>
> ### **1. Number of datasets**
>
> We agree that broader empirical coverage can be beneficial. However, our method is specifically designed for imbalanced classification, since the triplet loss and conditional sampling mechanism require discrete class labels. For this reason, we focus on classification datasets and carefully selected six benchmarks that span a wide spectrum of imbalance ratios (moderate to extreme), heterogeneous feature types (mixed categorical/numerical), and diverse real-world domains where minority-class performance is critical. With these datasets we cover many real-world use-cases (papers rarely use datasets with extreme imbalance ratios to evaluate their methods).
>
> Evaluating CTTVAE on regression or mixed tasks would not be meaningful, because the latent structuring and conditional components depend on discrete values (aka here the class labels) which is why our dataset selection reflects the scope of the method. TTVAE evaluates on a broad mixture of regression, density-estimation, and mixed-tabular tasks because it is designed as a general-purpose tabular generator.
>
> However, we are open to expanding the dataset number in the camera-ready version if appropriate.
>
>
> ### **2. Diffusion and foundation models**
>
> We agree that diffusion models are important baselines. In the revised version, we now include a diffusion-based model (TabDiff, ICLR 2025) as part of our evaluation.
>
> ### **3. 2D visualization**
>
> We added a 2D visualization in the appendix C (figure 5) that shows the performance comparison between our methods and the baselines.

---

### Official Review · Reviewer_yrzf · 2025-11-01

**Soundness:** 2
**Presentation:** 3
**Contribution:** 2
**Rating:** 4
**Confidence:** 3

**Summary:**

The paper proposes CTTVAE+TBS, a conditional transformer-based tabular variational autoencoder designed to generate high-quality synthetic tabular data in the presence of severe class imbalance. CTTVAE combines a class-aware triplet margin loss (which enforces compact and separable latent representations) with a training-by-sampling (TBS) strategy that adaptively increases exposure to underrepresented classes. This dual mechanism enables the model to produce synthetic data that better supports downstream tasks, particularly for minority categories, without sacrificing fidelity or privacy.

**Strengths:**

Generating synthetic datasets under severe class imbalance is a practically important problem with broad real-world relevance. The paper is well written and clearly organized, with careful optimization of baseline models and comprehensive details of the experimental setup and results provided in the appendix.

**Weaknesses:**

My main concern with the paper relate to its empirical evaluation. The paper compares CTTVAE only against GAN- and VAE-based models, which are now relatively weak baselines compared to modern diffusion-based approaches. The paper justify this by stating that diffusion models such as TabDDPM require significantly greater computational resources and are typically reported at the dataset level, making alignment with their evaluation setup impractical. However, this reasoning is not fully convincing. In practice, TabDDPM is not substantially more computationally demanding than models like TVAE or CTGAN. Indeed, when Wang and Nguyen (2025) introduced TTVAE, they included direct comparisons with both TabDDPM and TabSyn. It should therefore be feasible to run diffusion-based baselines on the same train/test splits, using the same number of replications, and to evaluate them with the same utility, fidelity, and privacy metrics—ensuring that hyperparameters are appropriately tuned (e.g., via Optuna). Including these stronger baselines would make the empirical analysis much more compelling.

Additionally, while line 56 indicates that CTTVAE is compared against two classical oversampling baselines, and Figure 4 in the Appendix lists both SMOTE and SMOTENC, the paper only reports results for SMOTE. The paper needs to clarify this point.

Other aspects of the paper that warrant improvement include:

1. The experiments are replicated only three times. Increasing the number of replications (e.g., to ten) would improve the statistical robustness of the results.

2. The paper evaluates only six datasets in its baseline comparisons. Expanding the number of benchmark datasets would strengthen the empirical evidence—for example, the TTVAE paper includes sixteen benchmarks.

3. In line 58, the paper claims to compare CTTVAE with five state-of-the-art baselines, but this statement should be moderated since models such as TVAE and CTGAN are no longer considered state-of-the-art.

4. Tables 3 and 4 should include standard deviations, and the main text should explicitly reference that per-dataset results are available in Tables 8–13 of the Appendix, which should likewise report standard deviations.

Minor issues:

The caption of Table 2 states that the table report results for both majority and minority groups, but only the minority group results are reported. (The paper might also want to point out that the results for both minority and majority classes are presented in Table 7 in the Appendix).

In line 123, the paper should cite Wang and Nguyen (2025) rather than Badaro et al. (2023).

The captions of Tables 3, 4, and 5 should define the abbreviations “Maj.” and “Min.” for clarity.

##########################################

Overall, this is paper that could be of relevance to the ICLR community. However, in its current form, I am inclined to recommend rejection due to significant limitations in the experimental evaluation—particularly the absence of comparisons with diffusion-based models. That said, I would be open to revisiting my assessment if the paper address these issues and provide evidence that the advantages of CTTVAE persist when evaluated against stronger diffusion-based baselines.

**Questions:**

Could you clarify why aligning the evaluations of diffusion models with the paper’s experimental setup is considered impractical?

Can you clarify why the paper did not pursue comparisons against SMOTENC?

The paper notes that DCR and NNDR are reported at the 5th percentile following Zhao et al. (2021), but the computation procedure is unclear. Is the 5th percentile taken over the distances between each synthetic record and all real records, or is DCR first computed per synthetic record and then the 5th percentile reported across all synthetic samples?

The paper omits details about the transformer architecture. Does this imply that the same configuration as in the TTVAE reference is used?

**Details Of Ethics Concerns:**

The README file accompanying the submitted code states under the Project Status section that the paper has been submitted to AAAI 2026. This is most likely a simple oversight—perhaps the authors initially planned to submit to AAAI and forgot to update the README. However, I wanted to flag this here in case it indicates a potential double submission.

---

> ### Author Response · Authors · 2025-11-25
> **Response to Reviewer yrzf**
>
> We thank the reviewer for the detailed and constructive feedback. Below, we respond to each of your remarks and questions in turn.
>
> ## **Addressing weaknesses:**
>
> ### **1. Number of replications**
>
> We followed the evaluation protocol used in CTABGAN, which also performs three independent generations per method.
>
> We agree that increasing the number of replications would further strengthen robustness, and we are open to expanding this in the next version if appropriate.
>
> ### **2. Number of datasets**
>
> We agree that broader empirical coverage can be beneficial. However, our method is specifically designed for imbalanced classification, since the triplet loss and conditional sampling mechanism require discrete class labels. For this reason, we focus on classification datasets and carefully selected six benchmarks that span a wide spectrum of imbalance ratios (moderate to extreme), heterogeneous feature types (mixed categorical/numerical), and diverse real-world domains where minority-class performance is critical. With these datasets we cover many real-world use-cases and fall within the range of the number classification benchmarks used in the baselines (papers rarely use datasets with extreme imbalance ratios to evaluate their methods).
>
> Evaluating CTTVAE on regression or mixed tasks would not be meaningful, because the latent structuring and conditional components depend on discrete values (aka here the class labels) which is why our dataset selection reflects the scope of the method. TTVAE evaluates on a broad mixture of regression, density-estimation, and mixed-tabular tasks because it is designed as a general-purpose tabular generator.
>
> However, we are open to expanding the number of classification datasets in the next version if appropriate.
>
> ### **3. Statement moderation**
>
> We agree with the reviewer and have moderated the statement at line 58 to better reflect the standing of classical baselines.
>
> ### **4. Standard deviations**
>
> We added standard deviations for Tables 3–4 and Tables 8–13, and we explicitly reference these tables in the main text (l.292; l.332; l.375 and l.385).
>
> ## **Questions:**
> ### **1. Diffusion model baseline**
>
> We added a diffusion-based baseline in the revised version, TabDiff (ICLR 2025), and evaluate it under the same utility, fidelity, and privacy metrics as all other methods.
>
> ### **2. SMOTENC**
>
> SMOTENC is designed specifically for datasets that contain both continuous and categorical features. However, one of our six datasets (CR) contains only numerical variables, for which SMOTENC is not defined.
>
> To maintain a consistent, uniform baseline suite across all six datasets, we chose to retain SMOTE (which is applicable to all datasets) and removed SMOTENC. Its earlier mention in the text was an oversight and has been corrected in the revision.
>
> ### **3. DCR/NNDR**
>
> The 5th percentile is taken over values computed per real sample, not per synthetic sample.
>
> In the implementation taken from the CTABGAN paper (Zhao et al, 2021), we treat the real data as the reference set. For each real sample, we compute its two closest synthetic neighbors $(d_1, d_2)$ after subsampling 15% of real and synthetic samples  and z-score normalization (appendix A.3).
>
> - **DCR** is the 5th percentile across real samples of the nearest-neighbor distances $(d_1)$.
> - **NNDR** is the 5th percentile across real samples of the ratios $(\frac{d_1}{d_2})$.
>
> So we first compute a value per real record (using distances to synthetic data), and then report the 5th percentile across those real sample values.
>
> ### **4. Architecture details**
>
> We follow the architectural components of TTVAE directly and do not modify the transformer backbone. We clarified this at line 154 in the revised version.
>
> ## **Addressing minor issues**
>
> We appreciate the reviewer pointing these out. The following corrections were made:
> - Line 292 now correctly references the table reporting both majority and minority results.
> - Captions for Tables 3–5 now define “Maj.” and “Min.”
> - The reference at line 123 has been corrected to Wang & Nguyen (2025).
>
> ### **Addressing note for ethics review**
>
> We appreciate the reviewer for noticing this oversight. The mention of a prior AAAI submission in the README was an outdated version from an early draft of the repository and did not reflect the status of this submission. We confirm the paper is not under review at any other venue, and the README has now been corrected.

---

### Official Review · Reviewer_xVLk · 2025-11-01

**Soundness:** 2
**Presentation:** 3
**Contribution:** 2
**Rating:** 2
**Confidence:** 3

**Summary:**

This paper proposes CTTVAE+TBS, a conditional transformer-based variational autoencoder framework designed for tabular data generation under severe class imbalance. The model introduces two main components: a triplet-margin loss that structures the latent space to enforce class separability, and a training-by-sampling strategy that probabilistically adjusts batch composition to mitigate class frequency bias during training. Together, these mechanisms aim to improve minority-class representation and enhance the downstream utility of generated data while maintaining fidelity and privacy. Experimental results across multiple real-world datasets show that CTTVAE+TBS achieves stronger minority-class F1 scores and better privacy metrics than existing tabular data generators such as SMOTE, CTGAN, and TTVAE, though its overall utility gains are sometimes inconsistent.

**Strengths:**

The paper is well written and clearly structured, making the overall methodology and experimental design easy to follow.

**Weaknesses:**

1. The paper lacks a clear explanation of how 'balanced datasets' are constructed in experiments, and it is also unclear how many minority samples were actually added or generated to compose the balanced dataset used for experiments.


2. In several datasets, models trained on the original imbalanced data outperform those trained on oversampled data, raising questions about the true benefit of the proposed oversampling process. If the primary motivation is to generate standalone synthetic data rather than to improve classifier performance, the paper should include a direct evaluation of the quality, representativeness, and practical utility of the generated samples themselves.


3. There exists diffusion-based oversampling methods, such as Sos (Score-based Oversampling for Tabular Data) [1], tackle the same imbalance problem through score-based generative modeling. The paper would benefit from a comparison or discussion that clarifies how CTTVAE+TBS differs from or complements these diffusion-based approaches.


[1] Kim, J., Lee, C., Shin, Y., Park, S., Kim, M., Park, N., & Cho, J. (2022, August). Sos: Score-based oversampling for tabular data. In Proceedings of the 28th ACM SIGKDD conference on knowledge discovery and data mining (pp. 762-772).

**Questions:**

1. What do you think might be the reason why oversampling sometimes performs worse than using the original data?

2. Could the authors clarify how the balanced dataset used in the experiments was composed?

**Details Of Ethics Concerns:**

no ethic concerns

---

> ### Author Response · Authors · 2025-11-25
> **Response to Reviewer xVLk**
>
> We thank the reviewer for the valuable feedback.
>
> We would like to clarify the scope and objective of our work.
>
> CTTVAE does not rebalance datasets. All generated datasets preserve the original class prior unless the user explicitly needs to do oversampling which can be done with its built-in conditional sampling capabilities. Our evaluation focuses on improving the quality and utility of minority samples under natural imbalance rather than on oversampling strategies.
>
>
> ## **Addressing weaknesses**
>
> ### **1. Regarding “balanced datasets” and oversampling motivation**
>
> No balanced synthetic datasets were created in this work. We focused on improving representation learning of minority samples for generative purposes without altering class priors. All experiments use fully synthetic datasets that match the original class distribution. Our goal is not to oversample or rebalance, but to improve the representativeness and utility of minority samples under the original priors. We clarified this more explicitly in the revised text (l.266).
>
> ### **2. Regarding why models trained on original data may outperform oversampled data**
>
> This observation does not apply to our setting, since we do not rebalance or oversample. We maintain the original class ratios and aim to improve minority quality (evaluated through utility, fidelity, and privacy metrics) without altering class frequencies.
>
> ### **3. Diffusion-based oversampling methods**
>
> We appreciate the reviewer’s suggestion regarding diffusion-based models. In the revised version, we now include TabDiff (ICLR 2025), a recent diffusion model designed specifically for tabular data. We evaluated it under the same utility, fidelity, and privacy metrics as the other baselines.
>
>
> ## **Addressing the reviewer's questions**
>
> ### **1. Interpretation of the comment regarding balanced vs. original dataset performance**
>
> Since we do not construct balanced datasets, we interpret this comment as referring to why fully synthetic datasets sometimes underperform real data. Minor variability of this kind can arise simply because the classifiers used in the MLE setup are trained separately on real and synthetic datasets but evaluated on the same real test set. Since synthetic data inevitably approximates but does not perfectly match the real distribution, classifiers trained on it may exhibit slightly lower performance. This is expected in generative modeling. However, the comparative trends between the generative models still are maintained.
>
> ### **2. Composition of the “balanced dataset”**
>
> Since we do not construct balanced datasets in any experiment, we understand this comment as referring to the composition of our synthetic datasets. All real dataset compositions used in our experiments (including class ratios) are provided in Table 4 for clarity and we generate synthetic datasets maintaining the same composition (l.266).

---

### Official Review · Reviewer_fUEk · 2025-11-05

**Soundness:** 3
**Presentation:** 3
**Contribution:** 2
**Rating:** 4
**Confidence:** 4

**Summary:**

The authors of the paper aim to tackle the problem of conditional generation of imbalanced data within tabular data, regime. They introduce a Conditional Transformer-based Tabular Variational Autoencoder (CTTVAE), an extension of TTVAE augmented by an additional loss, namely a class-aware triplet margin loss and a training-by-sampling (TBS) strategy that adaptively increases exposure to underrepresented groups. Through this adaptation they intend to restructure the latent space so that it better encapsulates the intra-class compactness and inter-class separation and to mitigate representation bias  when categorical features exhibit strong imbalance.
At the experiments section they explore the performance of their method against state of the arts methods, across six real-world benchmarks.

**Strengths:**

The authors of the paper introduce CTTVAE+TBS through adding a triplet loss to TTVAE and TBS. By doing this, they aim to conditionally generate imbalanced data, a task that is frequently overseen. They add a triplet margin loss to the TTVAEs loss function (that replaces KL divergence term, with Maximum Mean Discrepancy (MMD) penalty between the aggregated posterior q(z) and the Gaussian prior p(z)) in order to encourage embeddings of the same class to lie closer together than those of different classes.
Moreover they use a variant of the TBS concept, at which sampling is guided solely by a categorical variable as opposed to sampling over all discrete columns.

**Weaknesses:**

The method, albeit interesting, does not seem to significantly outperform existing methods. Actually SMOTE outperforms CTTVAE+TBS across all distance metrics.

three soft comments:
(1) there is a sum at equation (4) that is not explained (why, over what);
(2) at equation 6, it (most likely) should be $\hat{z}_i$;
(3) at Figure 1., above the circled set with the blue and red dots, it should probably be (z,h) instead of just z;

**Questions:**

I would like to ask the authors if they could please :

1) comment on the sensitivity of the method to hyperparemeters $\alpha$ and $\beta$ (Loss CTTVAE, page 4);
2) please explain what is the role of $u_r \sim \mathcal{U}(0,1)$, in equation 6.;
3) comment on the sensitivity and performance of the method with respect to the number of classes;
4) comment on the outperformance of SMOTE across all distance metrics (Table 3);
5) add the absolute difference between correlation matrices of SMOTE as well (Figure 3);
6) compare their results with [1], a diffusion model based method that also generates (and imputes) tabular data and also employ conditional generation


[1] Jolicoeur-Martineau, Alexia, Kilian Fatras, and Tal Kachman. "Generating and imputing tabular data via diffusion and flow-based gradient-boosted trees." International conference on artificial intelligence and statistics. PMLR, 2024.

---

> ### Author Response · Authors · 2025-11-25
> **Response to Reviewer fUEk**
>
> We thank the reviewer for the detailed and constructive comments. We address each point below.
>
> ## **Remarks**
>
> ### **1. On improvements over baselines**
>
> Our primary goal is to improve minority-class representation without degrading majority samples.
>
> CTTVAE yields consistent gains for minority-class F1 w.r.t. all baselines (Table 2), particularly on the highly imbalanced datasets where gains are significant (improves by 2.4 to 12.4 F1 points over TTVAE), while maintaining the same majority performance (Table 6).
>
> CTTVAE also provides a more balanced trade-off, preserving competitive fidelity while significantly improving privacy relative to interpolation-based sampling methods (SMOTE, TTVAE in latent space) and consistantly improving utility relative to all deep generative baselines as well as achieving better fidelity scores.
>
> ### **2. Equation (4)**
> The sum is taken over all valid triplets in the mini-batch. We updated the notation and added a clarification at line 170.
>
> ### **3. Sensitivity to multiclass datasets**
>
> We have not studied the method’s performance under a multiclass dataset. We focused solely on binary classification datasets. This could be extended in future version.
>
> ## **Questions/Comments**
>
> ### **1. Sensitivity to $\alpha$ and $\beta$**
>
> When $\alpha$ is too small, the latent space is only weakly structured by class, so minority-class clusters remain poorly separated and utility gains diminish (falling back to TTVAE's loss function). A higher $\beta$ promotes a more distentangled latent space and a lower one improves the reconstruction loss.
>
> We added these explanations in lines 182-185.
>
> ### **2. Role of $u_r \sim \mathcal{U}(0,1)$**
>
>  $u_r$ randomly scales the interpolation step toward a latent neighbor, producing latent samples throughout the local region rather than at a fixed point. This increases diversity while remaining within the class manifold. We added a clarification at line 205-206. This can be now visualized better in the improved version of figure 1.
>
> ### **4. SMOTE outperforming CTTVAE on some distance metrics**
>
> SMOTE performs better for distance metrics evaluating fidelity (WD and JSD, tables 3 and 8-10) but not for distance metrics evaluating privacy (DCR and NNDR, tables 4 and 11-13).
>
> SMOTE’s interpolation places samples near real points in the input space, producing strong fidelity but weak privacy (due to low nearest-neighbor distances NNDR/DCR), hence synthetic samples are too close to real ones and increase memorization risks.
>
> CTTVAE reduces this privacy gap while improving utility and keeping competitive fidelity. Interpolation-based sampling methods (SMOTE; TTVAE in latent space) tend to inherently score poorly on privacy metrics due to their sampling nature, however CTTVAE bridges this gap considerably with deep generative models and moves closer to the behavior of deep generative models in terms of privacy while retaining competitive fidelity.
>
> This trade-off is illustrated in Figure 5 (Appendix C), where privacy (NNDR/DCR) and fidelity (WD/JSD) are shown jointly against minority-class utility.
>
> ### **5. Absolute correlation differences**
>
> We added correlation difference results for SMOTE and TabDiff in Figure 2.
>
> ### **6. Diffusion baseline**
>
> We added TabDiff (ICLR 2025), a recent SOTA diffusion model as an additional baseline. We evaluated it under the same utility, fidelity, and privacy metrics as the other baselines.

---

### Author Response · Authors · 2025-11-25
**General Response to Reviewers**

We sincerely thank all reviewers for their detailed and constructive feedback. We have carefully revised the paper and addressed all comments. Below we summarize the key changes implemented in the updated version.

We will give detailed replies to each reviewer as well.

---

**Major Additions and Experimental Updates:**

•	**Added TabDiff baseline:** We now include TabDiff (ICLR 2025) as a diffusion-based comparison across all datasets. Hyperparameters tuning details are reported in the final table of the appendix.

---

**Clarifications and Structural Improvements:**

•	**Triplet mining:** The description of the semi-hard triplet mining algorithm has been moved to Appendix B, allowing more space for essential methodological explanations and results in the main text.

•	Minor phrasing corrections for clarity.

•	**Improved conditional generation illustration:** Figure 1 has been updated to more clearly depict the conditional generation mechanism with clearly separated and explained steps.

•	**Expanded correlation matrix comparison:** Figure 2 now includes additional baseline methods for a broader comparison.

•	**Visualization of model performances:** We added Figure 5 in Appendix C, providing a 2D representation of the model performances to see better the difference between each method.

•	**Logging results correction:** We identified a minor logging inconsistency originating from Table 11 that affected the displayed values in Tables 4 and 5. This was a presentation-only formatting oversight (the computations were correct and conclusions unchanged). The updated revision now reflects the accurate logged results.

•	**Appendix reorganization:** The appendix has been restructured to improve readability.

---

We hope these revisions address the reviewers’ concerns and clarify the paper’s contributions. We are grateful for the valuable feedback and would be happy to provide any additional clarification.

---

### Meta-Review · Area_Chair_9xhY · 2026-01-05

**Summary:**

The paper introduces CTTVAE+TBS, a framework designed to generate synthetic tabular data under conditions of severe class imbalance. The method combines a Conditional Transformer-based Tabular Variational Autoencoder with two core mechanisms: a class-aware triplet margin loss to restructure the latent space for better separation, and a Training-By-Sampling  strategy to increase exposure to minority groups. The authors demonstrate the method's performance across six real-world benchmarks, claiming improvements in downstream utility for minority classes while maintaining fidelity.

Despite the authors' active engagement during the rebuttal phase, significant concerns remain regarding the empirical validation and the overall contribution of the work. Reviewers consistently highlighted the limited experimental scope, noting that the evaluation on only six datasets is insufficient to robustly demonstrate the method's generalizability, especially when compared to related works that utilize much broader benchmarks. Furthermore, while the authors introduced a diffusion-based baseline during the revision to address criticisms regarding missing state-of-the-art comparisons, the proposed method still struggles to consistently outperform simpler baselines like SMOTE on fidelity metrics. Even if the discussion phase had been extended to allow for further exchanges, the fundamental issues regarding the insufficient breadth of the evaluation suite and the lack of a convincing performance advantage over established methods are unlikely to be fully resolved. Consequently, the paper in its current form does not meet the bar for acceptance at ICLR.

**Reviewer Concerns:**

As pointed out by Reviewer fUEk, the main advantage of the proposed method over SMOTE appears to lie in the privacy metrics. However, I am not sufficiently convinced that privacy is a critical issue within the specific problem setting considered in this paper. Consequently, I do not believe that this fundamental concern regarding the method's practical contribution compared to simple baselines could be effectively resolved through further discussion.

Multiple reviewers consistently requested the addition of diffusion-based baselines. Although they asked for a variety of methods, the authors only added TabDiff. Given that the performance of this specific method was not particularly outstanding, and considering the intent was to see a broader comparison with other diffusion-based baselines, I do not believe this concern has been fully resolved. Furthermore, relying on only six datasets for experimentation and validation remains a major issue; unfortunately, since no additional experiments were conducted to address this limitation, this concern also remains unresolved.

**Reviewer Scores:**

I do not believe that any of the reviewers would have been likely to raise their scores even if the discussion had continued.

---

### Decision · Program_Chairs · 2026-01-26

Reject